# Analysis of Variational Bayesian Latent Dirichlet Allocation: Weaker Sparsity than MAP

**Shinichi Nakajima**
Berlin Big Data Center, TU Berlin
Berlin 10587 Germany
nakajima@tu-berlin.de

**Issei Sato**
University of Tokyo
Tokyo 113-0033 Japan
sato@r.dl.itc.u-tokyo.ac.jp

**Masashi Sugiyama**
University of Tokyo
Tokyo 113-0033, Japan
sugi@k.u-tokyo.ac.jp

**Kazuho Watanabe**
Toyohashi University of Technology
Aichi 441-8580 Japan
wkazuho@cs.tut.ac.jp

**Hiroko Kobayashi**
Nikon Corporation
Kanagawa 244-8533 Japan
hiroko.kobayashi@nikon.com

## Abstract

*Latent Dirichlet allocation* (LDA) is a popular generative model of various objects such as texts and images, where an object is expressed as a mixture of latent *topics*. In this paper, we theoretically investigate *variational Bayesian* (VB) learning in LDA. More specifically, we analytically derive the leading term of the VB free energy under an asymptotic setup, and show that there exist transition thresholds in Dirichlet hyperparameters around which the sparsity-inducing behavior drastically changes. Then we further theoretically reveal the notable phenomenon that *VB tends to induce weaker sparsity than MAP* in the LDA model, which is opposed to other models. We experimentally demonstrate the practical validity of our asymptotic theory on real-world *Last.FM* music data.

## 1 Introduction

*Latent Dirichlet allocation* (LDA) [5] is a generative model successfully used in various applications such as text analysis [5], image analysis [15], genomics [6, 4], human activity analysis [12], and collaborative filtering [14, 20][1]. Given word occurrences of documents in a corpora, LDA expresses each document as a mixture of multinomial distributions, each of which is expected to capture a *topic*. The extracted topics provide bases in a low-dimensional feature space, in which each document is compactly represented. This topic expression was shown to be useful for solving various tasks including classification [15], retrieval [26], and recommendation [14].

Since rigorous Bayesian inference is computationally intractable in the LDA model, various approximation techniques such as *variational Bayesian* (VB) learning [3, 7] are used. Previous theoretical studies on VB learning revealed that VB tends to produce sparse solutions, e.g., in mixture models [24, 25, 13], hidden Markov models [11], Bayesian networks [23], and fully-observed matrix factorization [17]. Here, we mean by sparsity that VB exhibits the automatic relevance determination

(ARD) effect [19], which automatically prunes irrelevant degrees of freedom under non-informative or weakly sparse prior. Therefore, it is naturally expected that VB-LDA also produces a sparse solution (in terms of topics). However, it is often observed that VB-LDA does not generally give sparse solutions.

In this paper, we attempt to clarify this gap by theoretically investigating the sparsity-inducing mechanism of VB-LDA. More specifically, we first analytically derive the leading term of the VB free energy in some asymptotic limits, and show that there exist transition thresholds in Dirichlet hyperparameters around which the sparsity-inducing behavior changes drastically. We then analyze the behavior of MAP and its variants in a similar way, and show that *the VB solution is less sparse than the MAP solution* in the LDA model. This phenomenon is completely opposite to other models such as mixture models [24, 25, 13], hidden Markov models [11], Bayesian networks [23], and fully-observed matrix factorization [17], where VB tends to induce stronger sparsity than MAP. We numerically demonstrate the practical validity of our asymptotic theory using artificial and real-world *Last.FM* music data for collaborative filtering, and further discuss the peculiarity of the LDA model in terms of sparsity.

The free energy of VB-LDA was previously analyzed in [16], which evaluated the advantage of collapsed VB [21] over the original VB learning. However, that work focused on the difference between VB and collapsed VB, and neither the absolute free energy nor the sparsity was investigated. The update rules of VB was compared with those of MAP [2]. However, that work is based on approximation, and rigorous analysis was not made. To the best of our knowledge, our paper is the first work that theoretically elucidates the sparsity-inducing mechanism of VB-LDA.

## 2 Formulation

In this section, we introduce the latent Dirichlet allocation model and variational Bayesian learning.

### 2.1 Latent Dirichlet Allocation

Suppose that we observe $M$ documents, each of which consists of $N^{(m)}$ words. Each word is included in a vocabulary with size $L$. We assume that each word is associated with one of the $H$ topics, which is not observed. We express the word occurrence by an $L$-dimensional indicator vector $\boldsymbol{w}$, where one of the entries is equal to one and the others are equal to zero. Similarly, we express the topic occurrence as an $H$-dimensional indicator vector $\boldsymbol{z}$. We define the following functions that give the item numbers chosen by $\boldsymbol{w}$ and $\boldsymbol{z}$, respectively:

$$\acute{l}(\boldsymbol{w}) = l \text{ if } w_l = 1 \text{ and } w_{l'} = 0 \text{ for } l' \neq l, \qquad \acute{h}(\boldsymbol{z}) = h \text{ if } z_h = 1 \text{ and } z_{h'} = 0 \text{ for } h' \neq h.$$

In the latent Dirichlet allocation (LDA) model [5], the word occurrence $\boldsymbol{w}^{(n,m)}$ of the $n$-th position in the $m$-th document is assumed to follow the multinomial distribution:

$$p(\boldsymbol{w}^{(n,m)}|\boldsymbol{\Theta}, \boldsymbol{B}) = \prod_{l=1}^{L} \left( (\boldsymbol{B}\boldsymbol{\Theta}^{\top})_{l,m} \right)^{w_l^{(n,m)}} = (\boldsymbol{B}\boldsymbol{\Theta}^{\top})_{\acute{l}(\boldsymbol{w}^{(n,m)}),m}, \tag{1}$$

where $\boldsymbol{\Theta} \in [0,1]^{M \times H}$ and $\boldsymbol{B} \in [0,1]^{L \times H}$ are parameter matrices to be estimated. The rows of $\boldsymbol{\Theta}$ and the columns of $\boldsymbol{B}$ are probability mass vectors that sum up to one. We denote a column vector of a matrix by a bold lowercase letter, and a row vector by a bold lowercase letter with a tilde, i.e.,

$$\boldsymbol{\Theta} = (\boldsymbol{\theta}_1, \ldots, \boldsymbol{\theta}_H) = (\widetilde{\boldsymbol{\theta}}_1, \ldots, \widetilde{\boldsymbol{\theta}}_M)^{\top}, \qquad \boldsymbol{B} = (\boldsymbol{\beta}_1, \ldots, \boldsymbol{\beta}_H) = (\widetilde{\boldsymbol{\beta}}_1, \ldots, \widetilde{\boldsymbol{\beta}}_L)^{\top}.$$

With this notation, $\widetilde{\boldsymbol{\theta}}_m$ denotes the topic distribution of the $m$-th document, and $\boldsymbol{\beta}_h$ denotes the word distribution of the $h$-th topic.

Given the topic occurrence latent variable $\boldsymbol{z}^{(n,m)}$, the complete likelihood is written as

$$p(\boldsymbol{w}^{(n,m)}, \boldsymbol{z}^{(n,m)}|\boldsymbol{\Theta}, \boldsymbol{B}) = p(\boldsymbol{w}^{(n,m)}|\boldsymbol{z}^{(n,m)}, \boldsymbol{B})p(\boldsymbol{z}^{(n,m)}|\boldsymbol{\Theta}), \tag{2}$$

where $p(\boldsymbol{w}^{(n,m)}|\boldsymbol{z}^{(n,m)}, \boldsymbol{B}) = \prod_{l=1}^{L}\prod_{h=1}^{H}(B_{l,h})^{w_l^{(n,m)}z_h^{(n,m)}}$, $p(\boldsymbol{z}^{(n,m)}|\boldsymbol{\Theta}) = \prod_{h=1}^{H}(\Theta_{m,h})^{z_h^{(n,m)}}$.

We assume the Dirichlet prior on $\boldsymbol{\Theta}$ and $\boldsymbol{B}$:

$$p(\boldsymbol{\Theta}|\alpha) \propto \prod_{m=1}^{M}\prod_{h=1}^{H}(\Theta_{m,h})^{\alpha-1}, \qquad p(\boldsymbol{B}|\eta) \propto \prod_{h=1}^{H}\prod_{l=1}^{L}(B_{l,h})^{\eta-1}, \tag{3}$$

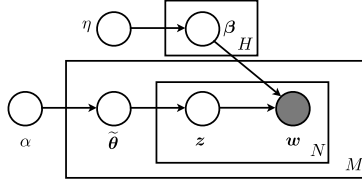

Figure 1: Graphical model of LDA.

where $\alpha$ and $\eta$ are hyperparameters that control the prior sparsity. We can make $\alpha$ dependent on $m$ and/or $h$, and $\eta$ dependent on $l$ and/or $h$, and they can be estimated from observation. However, we fix those hyperparameters as given constants for simplicity in our analysis below. Figure 1 shows the graphical model of LDA.

## 2.2 Variational Bayesian Learning

The Bayes posterior of LDA is written as

$$p(\boldsymbol{\Theta}, \boldsymbol{B}, \{\boldsymbol{z}^{(n,m)}\}|\{\boldsymbol{w}^{(n,m)}\}, \alpha, \eta) = \frac{p(\{\boldsymbol{w}^{(n,m)}\}, \{\boldsymbol{z}^{(n,m)}\}|\boldsymbol{\Theta}, \boldsymbol{B})p(\boldsymbol{\Theta}|\alpha)p(\boldsymbol{B}|\eta)}{p(\{\boldsymbol{w}^{(n,m)}\})}, \quad (4)$$

where $p(\{\boldsymbol{w}^{(n,m)}\}) = \int p(\{\boldsymbol{w}^{(n,m)}\}, \{\boldsymbol{z}^{(n,m)}\}|\boldsymbol{\Theta}, \boldsymbol{B})p(\boldsymbol{\Theta}|\alpha)p(\boldsymbol{B}|\eta)d\boldsymbol{\Theta}d\boldsymbol{B}d\{\boldsymbol{z}^{(n,m)}\}$ is intractable to compute and thus requires some approximation method. In this paper, we focus on the variational Bayesian (VB) approximation and investigate its behavior theoretically.

In the VB approximation, we assume that our approximate posterior is factorized as

$$q(\boldsymbol{\Theta}, \boldsymbol{B}, \{\boldsymbol{z}^{(n,m)}\}) = q(\boldsymbol{\Theta}, \boldsymbol{B})q(\{\boldsymbol{z}^{(n,m)}\}), \quad (5)$$

and minimize the free energy:

$$F = \left\langle \log \frac{q(\boldsymbol{\Theta}, \boldsymbol{B}, \{\boldsymbol{z}^{(n,m)}\})}{p(\{\boldsymbol{w}^{(n,m)}\}, \{\boldsymbol{z}^{(n,m)}\}|\boldsymbol{\Theta}, \boldsymbol{B})p(\boldsymbol{\Theta}|\alpha)p(\boldsymbol{B}|\eta)} \right\rangle_{q(\boldsymbol{\Theta}, \boldsymbol{B}, \{\boldsymbol{z}^{(n,m)}\})}, \quad (6)$$

where $\langle \cdot \rangle_p$ denotes the expectation over the distribution $p$. This amounts to finding the distribution that is closest to the Bayes posterior (4) under the constraint (5). Using the variational method, we can obtain the following stationary condition:

$$q(\boldsymbol{\Theta}) \propto p(\boldsymbol{\Theta}|\alpha) \exp \left\langle \log p(\{\boldsymbol{w}^{(n,m)}\}, \{\boldsymbol{z}^{(n,m)}\}|\boldsymbol{\Theta}, \boldsymbol{B}) \right\rangle_{q(\boldsymbol{B})q(\{\boldsymbol{z}^{(n,m)}\})}, \quad (7)$$

$$q(\boldsymbol{B}) \propto p(\boldsymbol{B}|\eta) \exp \left\langle \log p(\{\boldsymbol{w}^{(n,m)}\}, \{\boldsymbol{z}^{(n,m)}\}|\boldsymbol{\Theta}, \boldsymbol{B}) \right\rangle_{q(\boldsymbol{\Theta})q(\{\boldsymbol{z}^{(n,m)}\})}, \quad (8)$$

$$q(\{\boldsymbol{z}^{(n,m)}\}) \propto \exp \left\langle \log p(\{\boldsymbol{w}^{(n,m)}\}, \{\boldsymbol{z}^{(n,m)}\}|\boldsymbol{\Theta}, \boldsymbol{B}) \right\rangle_{q(\boldsymbol{\Theta})q(\boldsymbol{B})}. \quad (9)$$

From this, we can confirm that $\{q(\widetilde{\boldsymbol{\theta}}_m)\}$ and $\{q(\boldsymbol{\beta}_h)\}$ follow the Dirichlet distribution and $\{q(\boldsymbol{z}^{(n,m)})\}$ follows the multinomial distribution:

$$q(\boldsymbol{\Theta}) \propto \prod_{m=1}^{M} \prod_{h=1}^{H} (\Theta_{m,h})^{\breve{\Theta}_{m,h}-1}, \qquad q(\boldsymbol{B}) \propto \prod_{h=1}^{H} \prod_{l=1}^{L} (B_{l,h})^{\breve{B}_{l,h}-1}, \quad (10)$$

$$q(\{\boldsymbol{z}^{(n,m)}\}) = \prod_{m=1}^{M} \prod_{n=1}^{N^{(m)}} \prod_{h=1}^{H} (\widehat{z}_h^{(n,m)})^{z_h^{(n,m)}}, \quad (11)$$

where, for $\psi(\cdot)$ denoting the Digamma function, the variational parameters satisfy

$$\breve{\Theta}_{m,h} = \alpha + \sum_{n=1}^{N^{(m)}} \widehat{z}_h^{(n,m)}, \qquad \breve{B}_{l,h} = \eta + \sum_{m=1}^{M} \sum_{n=1}^{N^{(m)}} w_l^{(n,m)} \widehat{z}_h^{(n,m)}, \quad (12)$$

$$\widehat{z}_h^{(n,m)} = \frac{\exp\left\{\Psi(\breve{\Theta}_{m,h}) + \sum_{l=1}^{L} w_l^{(n,m)}\left(\Psi(\breve{B}_{l,h}) - \Psi\left(\sum_{l'=1}^{L} \breve{B}_{l',h}\right)\right)\right\}}{\sum_{h'=1}^{H} \exp\left\{\Psi(\breve{\Theta}_{m,h'}) + \sum_{l=1}^{L} w_l^{(n,m)}\left(\Psi(\breve{B}_{l,h'}) - \Psi\left(\sum_{l'=1}^{L} \breve{B}_{l',h'}\right)\right)\right\}}. \quad (13)$$

## 2.3 Partially Bayesian Learning and MAP Estimation

We can partially apply VB learning by approximating the posterior of $\boldsymbol{\Theta}$ or $\boldsymbol{B}$ by the delta function. This approach is called the *partially Bayesian* (PA) learning [18], whose behavior was analyzed

and compared with VB in fully-observed matrix factorization. We call it PBA learning if $\boldsymbol{\Theta}$ is marginalized and $\boldsymbol{B}$ is point-estimated, and PBB learning if $\boldsymbol{B}$ is marginalized and $\boldsymbol{\Theta}$ is point-estimated. Note that the original VB algorithm for LDA proposed by [5] corresponds to PBA in our terminology. We also analyze the behavior of MAP estimation, where both of $\boldsymbol{\Theta}$ and $\boldsymbol{B}$ are point-estimated. This corresponds to the *probabilistic latent semantic analysis* (pLSA) model [10], if we assume the flat prior $\alpha = \eta = 1$ [8].

## 3 Theoretical Analysis

In this section, we first give an explicit form of the free energy in the LDA model. We then investigate its asymptotic behavior for VB learning, and further conduct similar analyses to the PBA, PBB, and MAP methods. Finally, we discuss the sparsity-inducing mechanism of these learning methods, and the relation to previous theoretical studies.

### 3.1 Explicit Form of Free Energy

We first express the free energy (6) as a function of the variational parameters $\breve{\boldsymbol{\Theta}}$ and $\breve{\boldsymbol{B}}$:

$$F = R + Q, \qquad \text{where} \tag{14}$$

$$R = \left\langle \log \frac{q(\boldsymbol{\Theta})q(\boldsymbol{B})}{p(\boldsymbol{\Theta}|\alpha)p(\boldsymbol{B}|\eta)} \right\rangle_{q(\boldsymbol{\Theta},\boldsymbol{B})}$$

$$= \sum_{m=1}^{M} \left( \log \frac{\Gamma(\sum_{h=1}^{H}\breve{\Theta}_{m,h})}{\prod_{h=1}^{H}\Gamma(\breve{\Theta}_{m,h})} \frac{\Gamma(\alpha)^H}{\Gamma(H\alpha)} + \sum_{h=1}^{H} \left( \breve{\Theta}_{m,h} - \alpha \right) \left( \Psi(\breve{\Theta}_{m,h}) - \Psi(\sum_{h'=1}^{H}\breve{\Theta}_{m,h'}) \right) \right)$$

$$+ \sum_{h=1}^{H} \left( \log \frac{\Gamma(\sum_{l=1}^{L}\breve{B}_{l,h})}{\prod_{l=1}^{L}\Gamma(\breve{B}_{l,h})} \frac{\Gamma(\eta)^L}{\Gamma(L\eta)} + \sum_{l=1}^{L} \left( \breve{B}_{l,h} - \eta \right) \left( \Psi(\breve{B}_{l,h}) - \Psi(\sum_{l'=1}^{L}\breve{B}_{l',h}) \right) \right), \tag{15}$$

$$Q = \left\langle \log \frac{q(\{\boldsymbol{z}^{(n,m)}\})}{p(\{\boldsymbol{w}^{(n,m)}\},\{\boldsymbol{z}^{(n,m)}\}|\boldsymbol{\Theta},\boldsymbol{B})} \right\rangle_{q(\boldsymbol{\Theta},\boldsymbol{B},\{\boldsymbol{z}^{(n,m)}\})}$$

$$= -\sum_{m=1}^{M} N^{(m)} \sum_{l=1}^{L} V_{l,m} \log \left( \sum_{h=1}^{H} \frac{\exp(\Psi(\breve{\Theta}_{m,h}))}{\exp(\Psi(\sum_{h'=1}^{H}\breve{\Theta}_{m,h'}))} \frac{\exp(\Psi(\breve{B}_{l,h}))}{\exp(\Psi(\sum_{l'=1}^{L}\breve{B}_{l',h}))} \right). \tag{16}$$

Here, $\boldsymbol{V} \in \mathbb{R}^{L \times M}$ is the empirical word distribution matrix with its entries given by $V_{l,m} = \frac{1}{N^{(m)}} \sum_{n=1}^{N^{(m)}} w_l^{(n,m)}$. Note that we have eliminated the variational parameters $\{\widehat{\boldsymbol{z}}^{(n,m)}\}$ for the topic occurrence latent variables by using the stationary condition (13).

### 3.2 Asymptotic Analysis of VB Solution

Below, we investigate the leading term of the free energy in the asymptotic limit when $N \equiv \min_m N^{(m)} \to \infty$. Unlike the previous analysis for latent variable models [24], we do not assume $L, M \ll N$, but $1 \ll L, M, N$ at this point. This amounts to considering the asymptotic limit when $L, M, N \to \infty$ with a fixed mutual ratio, or equivalently, assuming $L, M \sim O(N)$. Throughout the paper, $H$ is set at $H = \min(L, M)$ (i.e., the matrix $\boldsymbol{B}\boldsymbol{\Theta}^\top$ can express any multinomial distribution). We assume that the word distribution matrix $\boldsymbol{V}$ is a sample from the multinomial distribution with the *true* parameter $\boldsymbol{U}^* \in \mathbb{R}^{L \times M}$ whose rank is $H^* \sim O(1)$, i.e., $\boldsymbol{U}^* = \boldsymbol{B}^* \boldsymbol{\Theta}^{*\top}$ where $\boldsymbol{\Theta}^* \in \mathbb{R}^{M \times H^*}$ and $\boldsymbol{B}^* \in \mathbb{R}^{L \times H^*}$.[2] We assume that $\alpha, \eta \sim O(1)$.

The stationary condition (12) leads to the following lemma (the proof is given in Appendix A):

**Lemma 1** *Let* $\widehat{\boldsymbol{B}}\widehat{\boldsymbol{\Theta}}^\top = \langle \boldsymbol{B}\boldsymbol{\Theta}^\top \rangle_{q(\boldsymbol{\Theta},\boldsymbol{B})}$. *Then, it holds that*

$$\langle (\boldsymbol{B}\boldsymbol{\Theta}^\top - \widehat{\boldsymbol{B}}\widehat{\boldsymbol{\Theta}}^\top)^2_{l,m} \rangle_{q(\boldsymbol{\Theta},\boldsymbol{B})} = O_p(N^{-2}), \tag{17}$$

$$Q = -\sum_{m=1}^{M} N^{(m)} \sum_{l=1}^{L} V_{l,m} \log(\widehat{\boldsymbol{B}}\widehat{\boldsymbol{\Theta}}^\top)_{l,m} + O_p(M), \tag{18}$$

*where* $O_p(\cdot)$ *denotes the order in probability.*

Eq.(17) implies the convergence of the posterior. Let

$$\widehat{J} = \sum_{l=1}^{L} \sum_{m=1}^{M} \kappa \left( (\widehat{\boldsymbol{B}}\widehat{\boldsymbol{\Theta}}^{\top})_{l,m} \neq (\boldsymbol{B}^*\boldsymbol{\Theta}^{*\top})_{l,m} + O_p(N^{-1}) \right) \tag{19}$$

be the number of entries of $\widehat{\boldsymbol{B}}\widehat{\boldsymbol{\Theta}}^{\top}$ that does not converge to the true value. Here, we denote by $\kappa(\cdot)$ the indicator function equal to one if the *event* is true, and zero otherwise. Then, Eq.(18) leads to the following lemma:

**Lemma 2** $Q$ is minimized when $\widehat{\boldsymbol{B}}\widehat{\boldsymbol{\Theta}}^{\top} = \boldsymbol{B}^*\boldsymbol{\Theta}^{*\top} + O_p(N^{-1})$, and it holds that

$$Q = S + O_p(\widehat{J}N + M), \qquad where$$

$$S = -\log p(\{\boldsymbol{w}^{(n,m)}\}, \{\boldsymbol{z}^{(n,m)}\}|\boldsymbol{\Theta}^*, \boldsymbol{B}^*) = -\sum_{m=1}^{M} N^{(m)} \sum_{l=1}^{L} V_{l,m} \log(\boldsymbol{B}^*\boldsymbol{\Theta}^*)_{l,m}.$$

Lemma 2 simply states that $Q/N$ converges to the normalized entropy $S/N$ of the true distribution (which is the lowest achievable value with probability 1), if and only if VB converges to the true distribution (i.e., $\widehat{J} = 0$).

Let $\widehat{H} = \sum_{h=1}^{H} \kappa(\frac{1}{M} \sum_{m=1}^{M} \widehat{\Theta}_{m,h} \sim O_p(1))$ be the number of topics used in the whole corpus, $\widehat{M}^{(h)} = \sum_{m=1}^{M} \kappa(\widehat{\Theta}_{m,h} \sim O_p(1))$ be the number of documents that contain the $h$-th topic, and $\widehat{L}^{(h)} = \sum_{l=1}^{L} \kappa(\widehat{B}_{l,h} \sim O_p(1))$ be the number of words of which the $h$-th topic consist. We have the following lemma (the proof is given in Appendix B):

**Lemma 3** $R$ is written as follows:

$$R = \left\{ M \left( H\alpha - \tfrac{1}{2} \right) + \widehat{H} \left( L\eta - \tfrac{1}{2} \right) - \sum_{h=1}^{\widehat{H}} \left( \widehat{M}^{(h)} \left( \alpha - \tfrac{1}{2} \right) + \widehat{L}^{(h)} \left( \eta - \tfrac{1}{2} \right) \right) \right\} \log N$$

$$+ (H - \widehat{H}) \left( L\eta - \tfrac{1}{2} \right) \log L + O_p(H(M+L)). \tag{20}$$

Since we assumed that the true matrices $\boldsymbol{\Theta}^*$ and $\boldsymbol{B}^*$ are of the rank of $H^*$, $\widehat{H} = H^* \sim O(1)$ is sufficient for the VB posterior to converge to the *true* distribution. However, $\widehat{H}$ can be much larger than $H^*$ with $\langle \boldsymbol{B}\boldsymbol{\Theta}^{\top} \rangle_{q(\boldsymbol{\Theta},\boldsymbol{B})}$ unchanged because of the non-identifiability of matrix factorization—duplicating topics with divided weights, for example, does not change the distribution.

Based on Lemma 2 and Lemma 3, we obtain the following theorem (the proof is given in Appendix C):

**Theorem 1** *In the limit when $N \to \infty$ with $L, M \sim O(1)$, it holds that $\widehat{J} = 0$ with probability 1, and*

$$F = S + \left\{ M \left( H\alpha - \tfrac{1}{2} \right) + \widehat{H} \left( L\eta - \tfrac{1}{2} \right) - \sum_{h=1}^{\widehat{H}} \left( \widehat{M}^{(h)} \left( \alpha - \tfrac{1}{2} \right) + \widehat{L}^{(h)} \left( \eta - \tfrac{1}{2} \right) \right) \right\} \log N$$

$$+ O_p(1).$$

*In the limit when $N, M \to \infty$ with $\frac{M}{N}, L \sim O(1)$, it holds that $\widehat{J} = o_p(\log N)$, and*

$$F = S + \left\{ M \left( H\alpha - \tfrac{1}{2} \right) - \sum_{h=1}^{\widehat{H}} \widehat{M}^{(h)} \left( \alpha - \tfrac{1}{2} \right) \right\} \log N + o_p(N \log N).$$

*In the limit when $N, L \to \infty$ with $\frac{L}{N}, M \sim O(1)$, it holds that $\widehat{J} = o_p(\log N)$, and*

$$F = S + HL\eta \log N + o_p(N \log N).$$

*In the limit when $N, L, M \to \infty$ with $\frac{L}{N}, \frac{M}{N} \sim O(1)$, it holds that $\widehat{J} = o_p(N \log N)$, and*

$$F = S + H(M\alpha + L\eta) \log N + o_p(N^2 \log N).$$

Since Eq.(17) was shown to hold, the predictive distribution converges to the true distribution if $\widehat{J} = 0$. Accordingly, Theorem 1 states that the consistency holds in the limit when $N \to \infty$ with $L, M \sim O(1)$.

Theorem 1 also implies that, in the asymptotic limits with small $L \sim O(1)$, the leading term depends on $\widehat{H}$, meaning that it dominates the topic sparsity of the VB solution. We have the following corollary (the proof is given in Appendix D):

Table 1: Sparsity thresholds of VB, PBA, PBB, and MAP methods (see Theorem 2). The first four columns show the thresholds $(\underline{\alpha}_{\text{sparse}}, \underline{\alpha}_{\text{dense}})$, of which the function forms depend on the range of $\eta$, in the limit when $N \to \infty$ with $L, M \sim O(1)$. A single value is shown if $\underline{\alpha}_{\text{sparse}} = \underline{\alpha}_{\text{dense}}$. The last column shows the threshold $\underline{\alpha}_{M\to\infty}$ in the limit when $N, M \to \infty$ with $\frac{M}{N}, L \sim O(1)$.

| $\eta$ range | $(\underline{\alpha}_{\text{sparse}}, \underline{\alpha}_{\text{dense}})$ | | | | $\underline{\alpha}_{M\to\infty}$ |
|---|---|---|---|---|---|
| | $0 < \eta \le \frac{1}{2L}$ | $\frac{1}{2L} < \eta \le \frac{1}{2}$ | $\frac{1}{2} < \eta < 1$ | $1 \le \eta < \infty$ | $0 < \eta < \infty$ |
| VB | $\frac{1}{2} - \frac{\frac{1}{2}-L\eta}{\min_h M^{*(h)}}$ | $\frac{1}{2} + \frac{L\eta-\frac{1}{2}}{\max_h M^{*(h)}}$ | $\left(\frac{1}{2} + \frac{L-1}{2\max_h M^{*(h)}}, \frac{1}{2} + \frac{L\eta-\frac{1}{2}}{\min_h M^{*(h)}}\right)$ | | $\frac{1}{2}$ |
| PBA | — | | | $\left(\frac{1}{2}, \frac{1}{2} + \frac{L(\eta-1)}{\min_h M^{*(h)}}\right)$ | $\frac{1}{2}$ |
| PBB | $1$ | $1 + \frac{L\eta-\frac{1}{2}}{\max_h M^{*(h)}}$ | $\left(1 + \frac{L-1}{2\max_h M^{*(h)}}, 1 + \frac{L\eta-\frac{1}{2}}{\min_h M^{*(h)}}\right)$ | | $1$ |
| MAP | — | | | $\left(1, 1 + \frac{L(\eta-1)}{\min_h M^{*(h)}}\right)$ | $1$ |

**Corollary 1** *Let* $M^{*(h)} = \sum_{m=1}^{M} \kappa(\Theta_{m,h}^* \sim O(1))$ *and* $L^{*(h)} = \sum_{l=1}^{L} \kappa(B_{l,h}^* \sim O(1))$. *Consider the limit when* $N \to \infty$ *with* $L, M \sim O(1)$. *When* $0 < \eta \le \frac{1}{2L}$, *the VB solution is sparse if* $\alpha < \frac{1}{2} - \frac{\frac{1}{2}-L\eta}{\min_h M^{*(h)}}$, *and dense if* $\alpha > \frac{1}{2} - \frac{\frac{1}{2}-L\eta}{\min_h M^{*(h)}}$. *When* $\frac{1}{2L} < \eta \le \frac{1}{2}$, *the VB solution is sparse if* $\alpha < \frac{1}{2} + \frac{L\eta-\frac{1}{2}}{\max_h M^{*(h)}}$, *and dense if* $\alpha > \frac{1}{2} + \frac{L\eta-\frac{1}{2}}{\max_h M^{*(h)}}$. *When* $\eta > \frac{1}{2}$, *the VB solution is sparse if* $\alpha < \frac{1}{2} + \frac{L-1}{2\max_h M^{*(h)}}$, *and dense if* $\alpha > \frac{1}{2} + \frac{L\eta-\frac{1}{2}}{\min_h M^{*(h)}}$. *In the limit when* $N, M \to \infty$ *with* $\frac{M}{N}, L \sim O(1)$, *the VB solution is sparse if* $\alpha < \frac{1}{2}$, *and dense if* $\alpha > \frac{1}{2}$.

In the case when $L, M \ll N$ and in the case when $L \ll M, N$, Corollary 1 provides information on the sparsity of the VB solution, which will be compared with other methods in Section 3.3. On the other hand, although we have successfully derived the leading term of the free energy also in the case when $M \ll L, N$ and in the case when $1 \ll L, M, N$, it unfortunately provides no information on sparsity of the solution.

### 3.3 Asymptotic Analysis of PBA, PBB, and MAP

By applying similar analysis to PBA learning, PBB learning, and MAP estimation, we can obtain the following theorem (the proof is given in Appendix E):

**Theorem 2** *In the limit when* $N \to \infty$ *with* $L, M \sim O(1)$, *the solution is sparse if* $\alpha < \underline{\alpha}_{\text{sparse}}$, *and dense if* $\alpha > \underline{\alpha}_{\text{dense}}$. *In the limit when* $N, M \to \infty$ *with* $\frac{M}{N}, L \sim O(1)$, *the solution is sparse if* $\alpha < \underline{\alpha}_{M\to\infty}$, *and dense if* $\alpha > \underline{\alpha}_{M\to\infty}$. *Here,* $\underline{\alpha}_{\text{sparse}}$, $\underline{\alpha}_{\text{dense}}$, *and* $\underline{\alpha}_{M\to\infty}$ *are given in Table 1.*

A notable finding from Table 1 is that the threshold that determines the topic sparsity of PBB-LDA is (most of the case exactly) $\frac{1}{2}$ larger than the threshold of VB-LDA. The same relation is observed between MAP-LDA and PBA-LDA. From these, we can conclude that point-estimating $\Theta$, instead of integrating it out, increases the threshold by $\frac{1}{2}$ in the LDA model. We will validate this observation by numerical experiments in Section 4.

### 3.4 Discussion

The above theoretical analysis (Thereom 2) showed that VB tends to induce weaker sparsity than MAP in the LDA model[3], i.e., VB requires sparser prior (smaller $\alpha$) than MAP to give a sparse solution (mean of the posterior). This phenomenon is completely opposite to other models such as mixture models [24, 25, 13], hidden Markov models [11], Bayesian networks [23], and fully-observed matrix factorization [17], where VB tends to induce stronger sparsity than MAP. This phenomenon might be partly explained as follows: In the case of mixture models, the sparsity threshold depends on the degree of freedom of a single component [24]. This is reasonable because

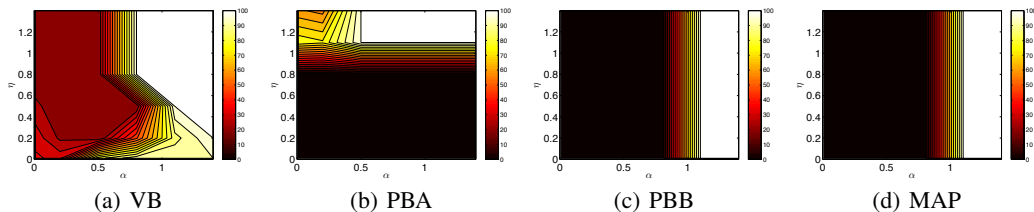

(a) VB  (b) PBA  (c) PBB  (d) MAP

Figure 2: Estimated number $\widehat{H}$ of topics by (a) VB, (b) PBA, (c) PBB, and (d) MAP, for the *artificial* data with $L = 100, M = 100, H^* = 20$, and $N \sim 10000$.

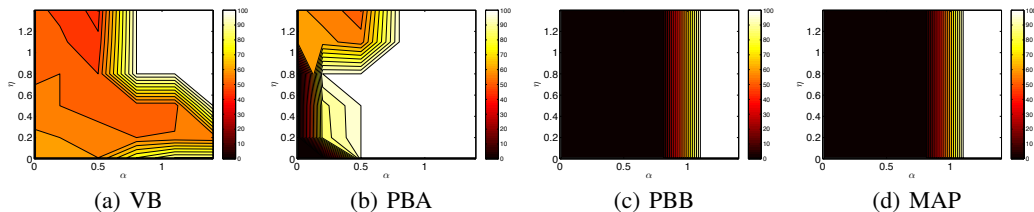

(a) VB  (b) PBA  (c) PBB  (d) MAP

Figure 3: Estimated number $\widehat{H}$ of topics for the *Last.FM* data with $L = 100, M = 100$, and $N \sim 700$.

adding a single component increases the model complexity by this amount. Also, in the case of LDA, adding a single topic requires additional $L + 1$ parameters. However, the added topic is shared over $M$ documents, which could discount the increased model complexity relative to the increased data fidelity. Corollary 1, which implies the dependency of the threshold for $\alpha$ on $L$ and $M$, might support this conjecture. However, the same applies to the matrix factorization, where VB was shown to give a sparser solution than MAP [17]. Investigation on related models, e.g., Poisson MF [9], would help us fully explain this phenomenon.

Technically, our theoretical analysis is based on the previous asymptotic studies on VB learning conducted for latent variable models [24, 25, 13, 11, 23]. However, our analysis is not just a straightforward extension of those works to the LDA model. For example, the previous analysis either implicitly [24] or explicitly [13] assumed the consistency of VB learning, while we also analyzed the consistency of VB-LDA, and showed that the consistency does not always hold (see Theorem 1). Moreover, we derived a general form of the asymptotic free energy, which can be applied to different asymptotic limits. Specifically, the standard asymptotic theory requires a large number $N$ of words per document, compared to the number $M$ of documents and the vocabulary size $L$. This may be reasonable in some collaborative filtering data such as the *Last.FM* data used in our experiments in Section 4. However, $L$ and/or $M$ would be comparable to or larger than $N$ in standard text analysis.

Our general form of the asymptotic free energy also allowed us to elucidate the behavior of the VB free energy when $L$ and/or $M$ diverges with the same order as $N$. This attempt successfully revealed the sparsity of the solution for the case when $M$ diverges while $L \sim O(1)$. However, when $L$ diverges, we found that the leading term of the free energy does not contain interesting insight into the sparsity of the solution. Higher-order asymptotic analysis will be necessary to further understand the sparsity-inducing mechanism of the LDA model with large vocabulary.

## 4    Numerical Illustration

In this section, we conduct numerical experiments on artificial and real data for collaborative filtering.

The *artificial* data were created as follows. We first sample the *true* document matrix $\boldsymbol{\Theta}^*$ of size $M \times H^*$ and the *true* topic matrix $\boldsymbol{B}^*$ of size $L \times H^*$. We assume that each row $\widetilde{\boldsymbol{\theta}}_m^*$ of $\boldsymbol{\Theta}^*$ follows the Dirichlet distribution with $\alpha^* = 1/H^*$, while each column $\boldsymbol{\beta}_h^*$ of $\boldsymbol{B}^*$ follows the Dirichlet distribution with $\eta^* = 1/L$. The document length $N^{(m)}$ is sampled from the Poisson distribution with its mean $N$. The word histogram $N^{(m)}\boldsymbol{v}_m$ for each document is sampled from the multinomial

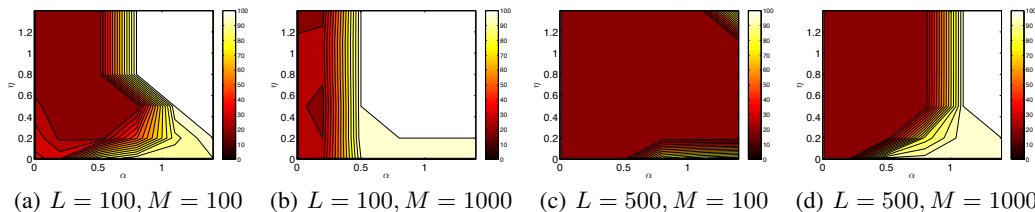

(a) $L = 100, M = 100$    (b) $L = 100, M = 1000$    (c) $L = 500, M = 100$    (d) $L = 500, M = 1000$

Figure 4: Estimated number $\widehat{H}$ of topics by VB-LDA for the *artificial* data with $H^* = 20$ and $N \sim 10000$. For the case when $L = 500, M = 1000$, the maximum estimated rank is limited to 100 for computational reason.

distribution with the parameter specified by the $m$-th row vector of $\boldsymbol{B}^* \boldsymbol{\Theta}^{*\top}$. Thus, we obtain the $L \times M$ matrix $\boldsymbol{V}$, which corresponds to the empirical word distribution over $M$ documents.

As a real-world dataset, we used the *Last.FM* dataset.[4] *Last.FM* is a well-known social music web site, and the dataset includes the triple ("user," "artist," "Freq") which was collected from the play-lists of users in the community by using a plug-in in users' media players. This triple means that "user" played "artist" music "Freq" times, which indicates users' preferred artists. A user and a played artist are analogous to a document and a word, respectively. We randomly chose $L$ artists from the top 1000 frequent artists, and $M$ users who live in the United States. To find a better local solution (which hopefully is close to the global solution), we adopted a split and merge strategy [22], and chose the local solution giving the lowest free energy among different initialization schemes.

Figure 2 shows the estimated number $\widehat{H}$ of topics by different approximation methods, i.e., VB, PBA, PBB, and MAP, for the *Artificial* data with $L = 100, M = 100, H^* = 20$, and $N \sim 10000$. We can clearly see that the sparsity threshold in PBB and MAP, where $\boldsymbol{\Theta}$ is point-estimated, is larger than that in VB and PBA, where $\boldsymbol{\Theta}$ is marginalized. This result supports the statement by Theorem 2. Figure 3 shows results on the *Last.FM* data with $L = 100, M = 100$ and $N \sim 700$. We see a similar tendency to Figure 2 except the region where $\eta < 1$ for PBA, in which our theory does not predict the estimated number of topics.

Finally, we investigate how different asymptotic settings affect the topic sparsity. Figure 4 shows the sparsity dependence on $L$ and $M$ for the *artificial* data. The graphs correspond to the four cases mentioned in Theorem 1, i.e, (a) $L, M \ll N$, (b) $L \ll N, M$, (c) $M \ll N, L$, and (d) $1 \ll N, L, M$. Corollary 1 explains the behavior in (a) and (b), and further analysis is required to explain the behavior in (c) and (d).

## 5   Conclusion

In this paper, we considered variational Bayesian (VB) learning in the latent Dirichlet allocation (LDA) model and analytically derived the leading term of the asymptotic free energy. When the vocabulary size is small, our result theoretically explains the phase-transition phenomenon. On the other hand, when vocabulary size is as large as the number of words per document, the leading term tells nothing about sparsity. We need more accurate analysis to clarify the sparsity in such cases.

Throughout the paper, we assumed that the hyperparameters $\alpha$ and $\eta$ are pre-fixed. However, $\alpha$ would often be estimated for each topic $h$, which is one of the advantages of using the LDA model in practice [5]. In the future work, we will extend the current line of analysis to the *empirical Bayesian* setting where the hyperparameters are also learned, and further elucidate the behavior of the LDA model.

**Acknowledgments**

The authors thank the reviewers for helpful comments. Shinichi Nakajima thanks the support from Nikon Corporation, MEXT Kakenhi 23120004, and the Berlin Big Data Center project (FKZ 01IS14013A). Masashi Sugiyama thanks the support from the JST CREST program. Kazuho Watanabe thanks the support from JSPS Kakenhi 23700175 and 25120014.

## Footnotes

[1] For simplicity, we use the terminology in text analysis below. However, the range of application of our theory given in this paper is not limited to texts.

[2] More precisely, $\boldsymbol{U}^* = \boldsymbol{B}^* \boldsymbol{\Theta}^{*\top} + O(N^{-1})$ is sufficient.

[3] Although this tendency was previously pointed out [2] by using the approximation $\exp(\psi(n)) \approx n - \frac{1}{2}$ and comparing the stationary condition, our result has first clarified the sparsity behavior of the solution based on the asymptotic free energy analysis without using such an approximation.

[4]http://mtg.upf.edu/node/1671

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
