[Supplementary Material · appendix_nips2014.pdf]

## A Proof of Lemma 1

The following bounds are known [1]:

$$\left(y - \frac{1}{2}\right)\log y - y + \frac{1}{2}\log(2\pi) \le \log \Gamma(y) \le \left(y - \frac{1}{2}\right)\log y - y + \frac{1}{2}\log(2\pi) + \frac{1}{12y}, \quad (21)$$

$$\log y - \frac{1}{y} \le \Psi(y) \le \log y - \frac{1}{2y}. \quad (22)$$

For the Dirichlet distribution $p(\boldsymbol{\theta}|\breve{\boldsymbol{\theta}}) \propto \prod_{h=1}^{H} a_h^{\breve{\theta}_h - 1}$, the mean and the variance are given as follows:

$$\widehat{\theta}_h = \langle \theta_h \rangle_{p(\boldsymbol{\theta}|\breve{\boldsymbol{\theta}})} = \frac{\breve{\theta}_h}{\breve{\theta}_0}, \qquad \langle (\theta_h - \widehat{\theta}_h)^2 \rangle_{p(\boldsymbol{\theta}|\breve{\boldsymbol{\theta}})} = \frac{\breve{\theta}_h(\breve{\theta}_0 - \breve{\theta}_h)}{\breve{\theta}_0^2(\breve{\theta}_0 + 1)},$$

where $\breve{\theta}_0 = \sum_{h=1}^{H} \breve{\theta}_h$.

For fixed $N$, $R$, defined by Eq.(15), diverges to $+\infty$ if $\breve{\Theta}_{m,h} \to +0$ for any $(m, h)$ or $\breve{B}_{l,h} \to +0$ for any $(l, h)$. Therefore, the global minimizer of the free energy (14) is in the interior of the domain, where the free energy is differentiable. Consequently, the global minimizer is a stationary point. The stationary condition (12) implies that

$$\breve{\Theta}_{m,h} \ge \alpha, \qquad\qquad\qquad \breve{B}_{l,h} \ge \eta, \qquad (23)$$

$$\sum_{h=1}^{H} \breve{\Theta}_{m,h} = \sum_{h=1}^{H} \alpha + N^{(m)}, \qquad \sum_{l=1}^{L} \breve{B}_{l,h} = \sum_{l=1}^{L} \eta + \sum_{m=1}^{M}(\breve{\Theta}_{m,h} - \alpha). \qquad (24)$$

Therefore, we have

$$\langle (\Theta_{m,h} - \widehat{\Theta}_{m,h})^2 \rangle_{q(\boldsymbol{\Theta})} = O_p(N^{-2}) \qquad \text{for all } (m, h), \qquad (25)$$

$$\left(\max_m \widehat{\Theta}_{m,h}\right)^2 \langle (B_{l,h} - \widehat{B}_{l,h})^2 \rangle_{q(\boldsymbol{B})} = O_p(N^{-2}) \qquad \text{for all } (l, h), \qquad (26)$$

which leads to Eq.(17).

By using Eq.(22), $Q$ is bounded as follows:

$$\underline{Q} \le Q \le \overline{Q},$$

where

$$\overline{Q} = -\sum_{m=1}^{M} N^{(m)} \sum_{l=1}^{L} V_{l,m} \log \left( \sum_{h=1}^{H} \frac{\breve{\Theta}_{m,h}}{\sum_{h'=1}^{H}\breve{\Theta}_{m,h'}} \frac{\breve{B}_{l,h}}{\sum_{l'=1}^{L}\breve{B}_{l',h}} \frac{\exp\left(-\frac{1}{\breve{\Theta}_{m,h}}\right)}{\exp\left(-\frac{1}{2\sum_{h'=1}^{H}\breve{\Theta}_{m,h'}}\right)} \frac{\exp\left(-\frac{1}{\breve{B}_{l,h}}\right)}{\exp\left(-\frac{1}{2\sum_{l'=1}^{L}\breve{B}_{l',h}}\right)} \right),$$

$$\underline{Q} = -\sum_{m=1}^{M} N^{(m)} \sum_{l=1}^{L} V_{l,m} \log \left( \sum_{h=1}^{H} \frac{\breve{\Theta}_{m,h}}{\sum_{h'=1}^{H}\breve{\Theta}_{m,h'}} \frac{\breve{B}_{l,h}}{\sum_{l'=1}^{L}\breve{B}_{l',h}} \frac{\exp\left(-\frac{1}{2\breve{\Theta}_{m,h}}\right)}{\exp\left(-\frac{1}{\sum_{h'=1}^{H}\breve{\Theta}_{m,h'}}\right)} \frac{\exp\left(-\frac{1}{2\breve{B}_{l,h}}\right)}{\exp\left(-\frac{1}{\sum_{l'=1}^{L}\breve{B}_{l',h}}\right)} \right).$$

Using Eqs.(25) and (26), we have Eq.(18), which completes the proof of Lemma 1. $\qquad \square$

## B Proof of Lemma 3

By using the bounds (21) and (22), $R$ can be bounded as

$$\underline{R} \le R \le \overline{R}, \qquad (27)$$

where

$$\underline{R} = -\sum_{m=1}^{M} \log\left(\frac{\Gamma(H\alpha)}{\Gamma(\alpha)^H}\right) - \sum_{h=1}^{H} \log\left(\frac{\Gamma(L\eta)^L}{\Gamma(\eta)}\right) - \frac{M(H-1) + H(L-1)}{2}\log(2\pi)$$

$$+ \sum_{m=1}^{M} \left\{ \left(H\alpha - \frac{1}{2}\right)\log\sum_{h=1}^{H}\breve{\Theta}_{m,h} - \sum_{h=1}^{H}\left(\alpha - \frac{1}{2}\right)\log\breve{\Theta}_{m,h} \right\}$$

$$+ \sum_{h=1}^{H} \left\{ \left( L\eta - \frac{1}{2} \right) \log \sum_{l=1}^{L} \breve{B}_{l,h} - \sum_{l=1}^{L} \left( \eta - \frac{1}{2} \right) \log \breve{B}_{l,h} \right\}$$

$$+ \sum_{m=1}^{M} \left\{ - \sum_{h=1}^{H} \frac{1}{12 \breve{\Theta}_{m,h}} - \sum_{h=1}^{H} \left( \breve{\Theta}_{m,h} - \alpha \right) \left( \frac{1}{\breve{\Theta}_{m,h}} - \frac{1}{2 \sum_{h'=1}^{H} \breve{\Theta}_{m,h'}} \right) \right\}$$

$$+ \sum_{h=1}^{H} \left\{ - \sum_{l=1}^{L} \frac{1}{12 \breve{B}_{l,h}} - \sum_{l=1}^{L} \left( \breve{B}_{l,h} - \eta \right) \left( \frac{1}{\breve{B}_{l,h}} - \frac{1}{2 \sum_{l'=1}^{L} \breve{B}_{l',h}} \right) \right\}, \tag{28}$$

$$\overline{R} = - \sum_{m=1}^{M} \log \left( \frac{\Gamma(H\alpha)}{\Gamma(\alpha)^H} \right) - \sum_{h=1}^{H} \log \left( \frac{\Gamma(L\eta)}{\Gamma(\eta)}^L \right) - \frac{M(H-1) + H(L-1)}{2} \log(2\pi)$$

$$+ \sum_{m=1}^{M} \left\{ \left( H\alpha - \frac{1}{2} \right) \log \sum_{h=1}^{H} \breve{\Theta}_{m,h} - \sum_{h=1}^{H} \left( \alpha - \frac{1}{2} \right) \log \breve{\Theta}_{m,h} \right\}$$

$$+ \sum_{h=1}^{H} \left\{ \left( L\eta - \frac{1}{2} \right) \log \sum_{l=1}^{L} \breve{B}_{l,h} - \sum_{l=1}^{L} \left( \eta - \frac{1}{2} \right) \log \breve{B}_{l,h} \right\}$$

$$+ \sum_{m=1}^{M} \left\{ \frac{1}{12 \sum_{h=1}^{H} \breve{\Theta}_{m,h}} - \sum_{h=1}^{H} \left( \breve{\Theta}_{m,h} - \alpha \right) \left( \frac{1}{2\breve{\Theta}_{m,h}} - \frac{1}{\sum_{h'=1}^{H} \breve{\Theta}_{m,h'}} \right) \right\}$$

$$+ \sum_{h=1}^{H} \left\{ \frac{1}{12 \sum_{l=1}^{L} \breve{B}_{l,h}} - \sum_{l=1}^{L} \left( \breve{B}_{l,h} - \eta \right) \left( \frac{1}{2\breve{B}_{l,h}} - \frac{1}{\sum_{l'=1}^{L} \breve{B}_{l',h}} \right) \right\}. \tag{29}$$

Eqs.(23) and (24) imply that

$$R = \sum_{m=1}^{M} \left\{ \left( H\alpha - \frac{1}{2} \right) \log \sum_{h=1}^{H} \breve{\Theta}_{m,h} - \sum_{h=1}^{H} \left( \alpha - \frac{1}{2} \right) \log \breve{\Theta}_{m,h} \right\}$$

$$+ \sum_{h=1}^{H} \left\{ \left( L\eta - \frac{1}{2} \right) \log \sum_{l=1}^{L} \breve{B}_{l,h} - \sum_{l=1}^{L} \left( \eta - \frac{1}{2} \right) \log \breve{B}_{l,h} \right\} + O_p(H(M+L)),$$

which leads to Eq.(20). This completes the proof of Lemma 3. $\qquad\square$

## C Proof of Theorem 1

Lemma 2 and Lemma 3 imply that the free energy can be written as follows:

$$F - S = \left\{ M \left( H\alpha - \tfrac{1}{2} \right) + \widehat{H} \left( L\eta - \tfrac{1}{2} \right) - \sum_{h=1}^{\widehat{H}} \left( \widehat{M}^{(h)} \left( \alpha - \tfrac{1}{2} \right) + \widehat{L}^{(h)} \left( \eta - \tfrac{1}{2} \right) \right) \right\} \log N$$

$$+ (H - \widehat{K}) \left( L\eta - \tfrac{1}{2} \right) \log L + O_p(\widehat{J}N + LM). \tag{30}$$

Below, we investigate the leading term of the free energy (30) in different asymptotic limits.

**In the limit when $N \to \infty$ with $L, M \sim O(1)$**

In this case, the minimizer should satisfy

$$\widehat{\boldsymbol{B}} \widehat{\boldsymbol{\Theta}}^{\top} = \boldsymbol{B}^* \boldsymbol{\Theta}^{*\top} + O_p(N^{-1}), \tag{31}$$

making $\widehat{J} = 0$ with probability 1, and the leading term of the order of $O_p(\log N)$:

$$F - S = \left\{ M \left( H\alpha - \tfrac{1}{2} \right) + \widehat{H} \left( L\eta - \tfrac{1}{2} \right) - \sum_{h=1}^{\widehat{H}} \left( \widehat{M}^{(h)} \left( \alpha - \tfrac{1}{2} \right) + \widehat{L}^{(h)} \left( \eta - \tfrac{1}{2} \right) \right) \right\} \log N + O_p(1).$$

Note that Eq.(31) implies the consistency of the predictive distribution.

**In the limit when $N, M \to \infty$ with $\frac{M}{N}, L \sim O(1)$**

In this case,

$$\widehat{J} = o_p(\log N), \tag{32}$$

making the leading term of the order of $O_p(N \log N)$:

$$F - S = \left\{ M \left( H\alpha - \tfrac{1}{2} \right) - \sum_{h=1}^{\widehat{H}} \widehat{M}^{(h)} \left( \alpha - \tfrac{1}{2} \right) \right\} \log N + o_p(N \log N).$$

Eq.(32) implies that the predictive distribution is not necessarily consistent—$o_p(\log N)$ components can deviate from the true matrix $\boldsymbol{B}^* \boldsymbol{\Theta}^{*\top}$ by the order of $O_p(1)$.

**In the limit when $N, L \to \infty$ with $\frac{L}{N}, M \sim O(1)$**

In this case, Eq.(32) holds, and the leading term of the free energy is of the order of $O_p(N \log N)$:

$$F - S = HL\eta \log N + o_p(N \log N).$$

**In the limit when $N, L, M \to \infty$ with $\frac{L}{N}, \frac{M}{N} \sim O(1)$**

In this case,

$$\widehat{J} = o_p(N \log N), \tag{33}$$

and the leading term of the free energy is of the order of $O_p(N^2 \log N)$:

$$F - S = H(M\alpha + L\eta) \log N + o_p(N^2 \log N).$$

This completes the proof of Theorem 1. □

## D  Proof of Corollary 1

From the compact representation when $\widehat{H} = H^*$, $\widehat{M}^{(h)} = M^{*(h)}$, and $\widehat{L}^{(h)} = L^{*(h)}$, we can decompose a singular component into two, keeping $\widehat{B}\widehat{\Theta}^\top$ unchanged, so that

$$\widehat{H} \to \widehat{H} + 1, \tag{34}$$

$$\sum_{h=1}^{H} \widehat{M}^{(h)} \to \sum_{h=1}^{H^*} \widehat{M}^{(h)} + \Delta M \qquad \text{for} \qquad \min_h M^{*(h)} \le \Delta M \le \max_h M^{*(h)}, \tag{35}$$

$$\sum_{h=1}^{H} \widehat{L}^{(h)} \to \sum_{h=1}^{H^*} \widehat{L}^{(h)} + \Delta L \qquad \text{for} \qquad 0 \le \Delta L \le \max_h L^{*(h)}. \tag{36}$$

Here the lower-bound for $\Delta M$ in Eq.(35) corresponds to the case that the least frequent topic is chosen to be decomposed, while the upper-bound to the case that the most frequent topic is chosen. The lower-bound for $\Delta L$ in Eq.(36) corresponds to the decomposition such that some of the word-occurrences are moved to a new topic, while the upper-bound to the decomposition such that the topic with the widest vocabulary is copied to a new topic. Note that the bounds both for $\Delta M$ and $\Delta L$ are not always achievable simultaneously, when we choose one topic to decompose.

Below, we investigate the relation between the sparsity of the solution and the hyperparameter setting in different asymptotic limits.

**In the limit when $N \to \infty$ with $L, M \sim O(1)$**

The coefficient of the leading term of the free energy is

$$\lambda = M\left(H\alpha - \tfrac{1}{2}\right) + \sum_{h=1}^{\widehat{H}}\left(L\eta - \tfrac{1}{2} - \widehat{M}^{(h)}\left(\alpha - \tfrac{1}{2}\right) - \widehat{L}^{(h)}\left(\eta - \tfrac{1}{2}\right)\right). \tag{37}$$

Note that the solution is sparse if Eq.(37) is increasing of $\widehat{H}$, and dense if it is decreasing. Eqs.(34)–(36) imply the following:

1. When $0 < \eta \le \frac{1}{2L}$ and $\alpha \le \frac{1}{2}$, the solution is sparse if

$$L\eta - \frac{1}{2} - \min_h M^{*(h)}\left(\alpha - \frac{1}{2}\right) > 0, \text{ or equivalently, } \alpha < \frac{1}{2} - \frac{1}{\min_h M^{*(h)}}\left(\frac{1}{2} - L\eta\right),$$

   and dense if

$$\alpha > \frac{1}{2} - \frac{1}{\min_h M^{*(h)}}\left(\frac{1}{2} - L\eta\right).$$

2. When $0 < \eta \le \frac{1}{2L}$ and $\alpha > \frac{1}{2}$, the solution is sparse if

$$L\eta - \frac{1}{2} - \max_h M^{*(h)}\left(\alpha - \frac{1}{2}\right) > 0, \text{ or equivalently, } \alpha < \frac{1}{2} - \frac{1}{\max_h M^{*(h)}}\left(\frac{1}{2} - L\eta\right),$$

   and dense if

$$\alpha > \frac{1}{2} - \frac{1}{\max_h M^{*(h)}}\left(\frac{1}{2} - L\eta\right).$$

   Therefore, the solution is always dense in this case.

3. When $\frac{1}{2L} < \eta \le \frac{1}{2}$ and $\alpha < \frac{1}{2}$, the solution is sparse if

$$L\eta - \frac{1}{2} - \min_h M^{*(h)} \left( \alpha - \frac{1}{2} \right) > 0, \text{ or equivalently, } \alpha < \frac{1}{2} + \frac{1}{\min_h M^{*(h)}} \left( L\eta - \frac{1}{2} \right),$$

and dense if

$$\alpha > \frac{1}{2} + \frac{1}{\min_h M^{*(h)}} \left( L\eta - \frac{1}{2} \right).$$

Therefore, the solution is always sparse in this case.

4. When $\frac{1}{2L} < \eta \le \frac{1}{2}$ and $\alpha \ge \frac{1}{2}$, the solution is sparse if

$$L\eta - \frac{1}{2} - \max_h M^{*(h)} \left( \alpha - \frac{1}{2} \right) > 0, \text{ or equivalently, } \alpha < \frac{1}{2} + \frac{1}{\max_h M^{*(h)}} \left( L\eta - \frac{1}{2} \right),$$

and dense if

$$\alpha > \frac{1}{2} + \frac{1}{\max_h M^{*(h)}} \left( L\eta - \frac{1}{2} \right).$$

5. When $\eta > \frac{1}{2}$ and $\alpha < \frac{1}{2}$, the solution is sparse if

$$L\eta - \frac{1}{2} - \max_h \left( M^{*(h)} \left( \alpha - \frac{1}{2} \right) + L^{*(h)} \left( \eta - \frac{1}{2} \right) \right) > 0, \tag{38}$$

and dense if

$$L\eta - \frac{1}{2} - \max_h \left( M^{*(h)} \left( \alpha - \frac{1}{2} \right) + L^{*(h)} \left( \eta - \frac{1}{2} \right) \right) < 0. \tag{39}$$

Therefore, the solution is sparse if

$$L\eta - \frac{1}{2} - \min_h M^{*(h)} \left( \alpha - \frac{1}{2} \right) - \max_h L^{*(h)} \left( \eta - \frac{1}{2} \right) > 0,$$

or equivalently, $\alpha < \frac{1}{2} + \frac{1}{\min_h M^{*(h)}} \left( L\eta - \frac{1}{2} - \max_h L^{*(h)} \left( \eta - \frac{1}{2} \right) \right),$

and dense if

$$L\eta - \frac{1}{2} - \max_h M^{*(h)} \left( \alpha - \frac{1}{2} \right) - \max_h L^{*(h)} \left( \eta - \frac{1}{2} \right) < 0,$$

or equivalently, $\alpha > \frac{1}{2} + \frac{1}{\max_h M^{*(h)}} \left( L\eta - \frac{1}{2} - \max_h L^{*(h)} \left( \eta - \frac{1}{2} \right) \right).$

Therefore, the solution is always sparse in this case.

6. When $\eta > \frac{1}{2}$ and $\alpha \ge \frac{1}{2}$, the solution is sparse if Eq.(38) holds, and dense if Eq.(39) holds. Therefore, the solution is sparse if

$$L\eta - \frac{1}{2} - \max_h M^{*(h)} \left( \alpha - \frac{1}{2} \right) - \max_h L^{*(h)} \left( \eta - \frac{1}{2} \right) > 0,$$

or equivalently, $\alpha < \frac{1}{2} + \frac{1}{\max_h M^{*(h)}} \left( L\eta - \frac{1}{2} - \max_h L^{*(h)} \left( \eta - \frac{1}{2} \right) \right),$

and dense if

$$L\eta - \frac{1}{2} - \min_h M^{*(h)} \left( \alpha - \frac{1}{2} \right) - \max_h L^{*(h)} \left( \eta - \frac{1}{2} \right) < 0,$$

or equivalently, $\alpha > \frac{1}{2} + \frac{1}{\min_h M^{*(h)}} \left( L\eta - \frac{1}{2} - \max_h L^{*(h)} \left( \eta - \frac{1}{2} \right) \right).$

Thus, we can conclude that, in this case, the solution is sparse if

$$\alpha < \frac{1}{2} + \frac{L - 1}{2 \max_h M^{*(h)}},$$

and dense if

$$\alpha > \frac{1}{2} + \frac{L\eta - \frac{1}{2}}{\min_h M^{*(h)}}.$$

Summarizing the above, we have the following lemma:

**Lemma 4** *When $0 < \eta \le \frac{1}{2L}$, the solution is sparse if $\alpha < \frac{1}{2} - \frac{\frac{1}{2} - L\eta}{\min_h M^{*(h)}}$, and dense if $\alpha > \frac{1}{2} - \frac{\frac{1}{2} - L\eta}{\min_h M^{*(h)}}$.*
*When $\frac{1}{2L} < \eta \le \frac{1}{2}$, the solution is sparse if $\alpha < \frac{1}{2} + \frac{L\eta - \frac{1}{2}}{\max_h M^{*(h)}}$, and dense if $\alpha > \frac{1}{2} + \frac{L\eta - \frac{1}{2}}{\max_h M^{*(h)}}$. When*
*$\eta > \frac{1}{2}$, the solution is sparse if $\alpha < \frac{1}{2} + \frac{L - 1}{2 \max_h M^{*(h)}}$, and dense if $\alpha > \frac{1}{2} + \frac{L\eta - \frac{1}{2}}{\min_h M^{*(h)}}$.*

**In the limit when** $N, M \to \infty$ **with** $\frac{M}{N}, L \sim O(1)$

The coefficient of the leading term of the free energy is given by

$$\lambda = M\left(H\alpha - \tfrac{1}{2}\right) - \sum_{h=1}^{\widehat{H}} \widehat{M}^{(h)}\left(\alpha - \tfrac{1}{2}\right). \tag{40}$$

Although the predictive distribution does not necessarily converges to the true distribution, we can investigate the sparsity of the solution by considering the duplication rules (34)–(36) that keep $\widehat{B}\widehat{\boldsymbol{\Theta}}^{\top}$ unchanged. It is clear that Eq.(40) is increasing of $\widehat{H}$ if $\alpha < \tfrac{1}{2}$, and decreasing if $\alpha > \tfrac{1}{2}$. Combing this result with Lemma 4 completes the proof of Corollary 1. $\qquad\square$

## E  Proof of Theorem 2

We analyze PBA learning, PBB learning, and MAP estimation, and then summarize the results.

### E.1  PBA Learning

The free energy for PBA learning is given as follows:

$$F^{\mathrm{PBA}} = \chi_B + R^{\mathrm{PBA}} + Q^{\mathrm{PBA}}, \tag{41}$$

where $\chi_B$ is a large constant corresponding to the negative entropy of the delta functions, and

$$R^{\mathrm{PBA}} = \left\langle \log \frac{q(\boldsymbol{\Theta})q(\boldsymbol{B})}{p(\boldsymbol{\Theta}|\alpha)p(\boldsymbol{B}|\eta)} \right\rangle_{q^{\mathrm{PBA}}(\boldsymbol{\Theta},\boldsymbol{B})}$$

$$= \sum_{m=1}^{M} \left( \log \frac{\Gamma(\sum_{h=1}^{H}\breve{\Theta}_{m,h}^{\mathrm{PBA}})}{\prod_{h=1}^{H}\Gamma(\breve{\Theta}_{m,h}^{\mathrm{PBA}})} \frac{\Gamma(\alpha)^H}{\Gamma(H\alpha)} + \sum_{h=1}^{H}\left(\breve{\Theta}_{m,h}^{\mathrm{PBA}} - \alpha\right)\left(\Psi(\breve{\Theta}_{m,h}^{\mathrm{PBA}}) - \Psi(\sum_{h'=1}^{H}\breve{\Theta}_{m,h'}^{\mathrm{PBA}})\right)\right)$$

$$+ \sum_{h=1}^{H}\left(\log \frac{\Gamma(\eta)^L}{\Gamma(L\eta)} + \sum_{l=1}^{L}(1-\eta)\left(\log(\breve{B}_{l,h}^{\mathrm{PBA}}) - \log(\sum_{l'=1}^{L}\breve{B}_{l',h}^{\mathrm{PBA}})\right)\right), \tag{42}$$

$$Q^{\mathrm{PBA}} = \left\langle \log \frac{q(\{\boldsymbol{z}^{(n,m)}\})}{p(\{\boldsymbol{w}^{(n,m)}\},\{\boldsymbol{z}^{(n,m)}\}|\boldsymbol{\Theta},\boldsymbol{B})} \right\rangle_{q^{\mathrm{PBA}}(\boldsymbol{\Theta},\boldsymbol{B},\{\boldsymbol{z}^{(n,m)}\})}$$

$$= -\sum_{m=1}^{M} N^{(m)} \sum_{l=1}^{L} V_{l,m} \log \left( \sum_{h=1}^{H} \frac{\exp\left(\Psi(\breve{\Theta}_{m,h}^{\mathrm{PBA}})\right)}{\exp\left(\Psi(\sum_{h'=1}^{H}\breve{\Theta}_{m,h'}^{\mathrm{PBA}})\right)} \frac{\breve{B}_{l,h}^{\mathrm{PBA}}}{\sum_{l'=1}^{L}\breve{B}_{l',h}^{\mathrm{PBA}}} \right). \tag{43}$$

Let us first consider the case when $\eta < 1$. In this case, $F$ diverges to $F \to -\infty$ with fixed $N$, when $\breve{B}_{l,h} = O(1)$ for any $(l,h)$ and $\breve{B}_{l',h} \to +0$ for all other $l' \neq l$. Therefore, the solution is useless.

When $\eta \geq 1$, the solution satisfies the following stationary condition:

$$\breve{\Theta}_{m,h}^{\mathrm{PBA}} = \alpha + \sum_{n=1}^{N^{(m)}} \widehat{z}_h^{\mathrm{PBA}(n,m)}, \qquad \breve{B}_{l,h}^{\mathrm{PBA}} = \eta - 1 + \sum_{m=1}^{M}\sum_{n=1}^{N^{(m)}} w_l^{(n,m)}\widehat{z}_h^{\mathrm{PBA}(n,m)}, \tag{44}$$

$$\widehat{z}_h^{\mathrm{PBA}(n,m)} = \frac{\exp\left(\Psi(\breve{\Theta}_{m,h}^{\mathrm{PBA}})\right)\prod_{l=1}^{L}(\breve{B}_{l,h}^{\mathrm{PBA}})^{w_l^{(n,m)}}}{\sum_{h'=1}^{H}\left(\exp\left(\Psi(\breve{\Theta}_{m,h'}^{\mathrm{PBA}})\right)\prod_{l=1}^{L}(\breve{B}_{l,h'}^{\mathrm{PBA}})^{w_l^{(n,m)}}\right)}. \tag{45}$$

In the same way as for VB learning, we can obtain the following lemma:

**Lemma 5** *Let* $\widehat{\boldsymbol{B}}^{\mathrm{PBA}}\widehat{\boldsymbol{\Theta}}^{\mathrm{PBA}\top} = \langle \boldsymbol{B}\boldsymbol{\Theta}^{\top}\rangle_{q^{\mathrm{PBA}}(\boldsymbol{\Theta},\boldsymbol{B})}$. *Then, it holds that*

$$\langle(\boldsymbol{B}\boldsymbol{\Theta}^{\top} - \widehat{\boldsymbol{B}}^{\mathrm{PBA}}\widehat{\boldsymbol{\Theta}}^{\mathrm{PBA}\top})_{l,m}^2\rangle_{q^{\mathrm{PBA}}(\boldsymbol{\Theta},\boldsymbol{B})} = O_p(N^{-2}), \tag{46}$$

$$Q^{\mathrm{PBA}} = -\sum_{m=1}^{M} N^{(m)} \sum_{l=1}^{L} V_{l,m} \log(\widehat{\boldsymbol{B}}^{\mathrm{PBA}}\widehat{\boldsymbol{\Theta}}^{\mathrm{PBA}\top})_{l,m} + O_p(N^{-1}). \tag{47}$$

$Q^{\mathrm{PBA}}$ *is minimized when* $\widehat{\boldsymbol{B}}^{\mathrm{PBA}}\widehat{\boldsymbol{\Theta}}^{\mathrm{PBA}\top} = \boldsymbol{B}^*\boldsymbol{\Theta}^{*\top} + O_p(N^{-1})$, *and it holds that*

$$Q^{\mathrm{PBA}} = S + O_p(\widehat{J}N + LM).$$

$R^{\mathrm{PBA}}$ *is written as follows:*

$$R^{\mathrm{PBA}} = \left\{ M\left(H\alpha - \tfrac{1}{2}\right) + \widehat{H}L(\eta-1) - \sum_{h=1}^{\widehat{H}}\left(\widehat{M}^{(h)}\left(\alpha - \tfrac{1}{2}\right) + \widehat{L}^{(h)}(\eta-1)\right)\right\}\log N$$

$$+ (H - \widehat{H})L(\eta-1)\log L + O_p(H(M+L)). \tag{48}$$

Taking the different asymptotic limits, we obtain the following theorem:

**Theorem 3** *When $\eta < 1$, each column vector of $\widehat{B}^{\text{PBA}}$ has only one non-zero entry. Assume below that $\eta \geq 1$. In the limit when $N \to \infty$ with $L, M \sim O(1)$, it holds that $\widehat{J} = 0$ with probability 1, and*

$$F^{\text{PBA}} = S + \left\{ M\left(H\alpha - \tfrac{1}{2}\right) + \widehat{H}L\left(\eta - 1\right) - \sum_{h=1}^{\widehat{H}} \left(\widehat{M}^{(h)}\left(\alpha - \tfrac{1}{2}\right) + \widehat{L}^{(h)}\left(\eta - 1\right)\right) \right\} \log N$$
$$+ O_p(1).$$

*In the limit when $N, M \to \infty$ with $\frac{M}{N}, L \sim O(1)$, it holds that $\widehat{J} = o_p(\log N)$, and*

$$F^{\text{PBA}} = S + \left\{ M\left(H\alpha - \tfrac{1}{2}\right) - \sum_{h=1}^{\widehat{H}} \widehat{M}^{(h)}\left(\alpha - \tfrac{1}{2}\right) \right\} \log N + o_p(N \log N).$$

*In the limit when $N, L \to \infty$ with $\frac{L}{N}, M \sim O(1)$, it holds that $\widehat{J} = o_p(\log N)$, and*

$$F^{\text{PBA}} = S + HL(\eta - 1)\log N + o_p(N \log N).$$

*In the limit when $N, L, M \to \infty$ with $\frac{L}{N}, \frac{M}{N} \sim O(1)$, it holds that $\widehat{J} = o_p(N \log N)$, and*

$$F^{\text{PBA}} = S + H(M\alpha + L(\eta - 1))\log N + o_p(N^2 \log N).$$

Note that Theorem 3 provides no information on the sparsity of the PBA solution for $\eta < 1$. Below, we investigate the sparsity of the solution for $\eta \geq 1$.

**In the limit when $N \to \infty$ with $L, M \sim O(1)$**

The coefficient of the leading term of the free energy is

$$\lambda^{\text{PBA}} = M\left(H\alpha - \tfrac{1}{2}\right) + \sum_{h=1}^{\widehat{H}} \left(L(\eta - 1) - \widehat{M}^{(h)}\left(\alpha - \tfrac{1}{2}\right) - \widehat{L}^{(h)}\left(\eta - 1\right)\right).$$

The solution is sparse if $\lambda^{\text{PBA}}$ is increasing of $\widehat{H}$, and dense if it is decreasing. We focus on the case when $\eta \geq 1$. Eqs.(34)–(36) imply the following:

1. When $\alpha < \frac{1}{2}$, the solution is sparse if

$$L(\eta - 1) - \max_h \left( M^{*(h)}\left(\alpha - \frac{1}{2}\right) + L^{*(h)}\left(\eta - 1\right) \right) > 0, \tag{49}$$

and dense if

$$L(\eta - 1) - \max_h \left( M^{*(h)}\left(\alpha - \frac{1}{2}\right) + L^{*(h)}\left(\eta - 1\right) \right) < 0. \tag{50}$$

Therefore, the solution is sparse if

$$L(\eta - 1) - \min_h M^{*(h)}\left(\alpha - \frac{1}{2}\right) - \max_h L^{*(h)}\left(\eta - 1\right) > 0,$$

or equivalently, $\alpha < \dfrac{1}{2} + \dfrac{\left(L - \max_h L^{*(h)}\right)(\eta - 1)}{\min_h M^{*(h)}},$

and dense if

$$L(\eta - 1) - \max_h M^{*(h)}\left(\alpha - \frac{1}{2}\right) - \max_h L^{*(h)}\left(\eta - 1\right) < 0,$$

or equivalently, $\alpha > \dfrac{1}{2} + \dfrac{\left(L - \max_h L^{*(h)}\right)(\eta - 1)}{\max_h M^{*(h)}}.$

Therefore, the solution is always sparse in this case.

2. When $\alpha \geq \frac{1}{2}$, the solution is sparse if Eq.(49) holds, and dense if Eq.(50) holds. Therefore, the solution is sparse if

$$L(\eta - 1) - \max_h M^{*(h)} \left( \alpha - \frac{1}{2} \right) - \max_h L^{*(h)} (\eta - 1) > 0,$$

or equivalently, $\alpha < \frac{1}{2} + \dfrac{\left( L - \max_h L^{*(h)} \right) (\eta - 1)}{\max_h M^{*(h)}},$

and dense if

$$L(\eta - 1) - \min_h M^{*(h)} \left( \alpha - \frac{1}{2} \right) - \max_h L^{*(h)} (\eta - 1) < 0,$$

or equivalently, $\alpha > \frac{1}{2} + \dfrac{\left( L - \max_h L^{*(h)} \right) (\eta - 1)}{\min_h M^{*(h)}}.$

Thus, we can conclude that, in this case, the solution is sparse if

$$\alpha < \frac{1}{2},$$

and dense if

$$\alpha > \frac{1}{2} + \frac{L(\eta - 1)}{\min_h M^{*(h)}}.$$

Summarizing the above, we have the following lemma:

**Lemma 6** *Assume that $\eta \geq 1$. The solution is sparse if $\alpha < \frac{1}{2}$, and dense if $\alpha > \frac{1}{2} + \frac{L(\eta-1)}{\min_h M^{*(h)}}$.*

**In the limit when $N, M \to \infty$ with $\frac{M}{N}, L \sim O(1)$**

The coefficient of the leading term of the free energy is given by

$$\lambda = M \left( H\alpha - \frac{1}{2} \right) - \sum_{h=1}^{\widehat{H}} \widehat{M}^{(h)} \left( \alpha - \frac{1}{2} \right). \tag{51}$$

Although the predictive distribution does not necessarily converges to the true distribution, we can investigate the sparsity of the solution by considering the duplication rules (34)–(36) that keep $\widehat{B}\widehat{\Theta}^{\top}$ unchanged. It is clear that Eq.(51) is increasing of $\widehat{H}$ if $\alpha < \frac{1}{2}$, and decreasing if $\alpha > \frac{1}{2}$. Combing this result with Lemma 6, we obtain the following corollary:

**Corollary 2** *Assume that $\eta \geq 1$. In the limit when $N \to \infty$ with $L, M \sim O(1)$, the PBA solution is sparse if $\alpha < \frac{1}{2}$, and dense if $\alpha > \frac{1}{2} + \frac{L(\eta-1)}{\min_h M^{*(h)}}$. In the limit when $N, M \to \infty$ with $\frac{M}{N}, L \sim O(1)$, the PBA solution is sparse if $\alpha < \frac{1}{2}$, and dense if $\alpha > \frac{1}{2}$.*

## E.2 PBB Learning

The free energy for PBB learning is given as follows:

$$F^{\mathrm{PBB}} = \chi_{\Theta} + R^{\mathrm{PBB}} + Q^{\mathrm{PBB}}, \tag{52}$$

where $\chi_{\Theta}$ is a large constant corresponding to the negative entropy of the delta functions, and

$$
\begin{aligned}
R^{\mathrm{PBB}} &= \left\langle \log \frac{q(\Theta)q(B)}{p(\Theta|\alpha)p(B|\eta)} \right\rangle_{q^{\mathrm{PBB}}(\Theta, B)} \\
&= \sum_{m=1}^{M} \left( \log \frac{\Gamma(\alpha)^H}{\Gamma(H\alpha)} + \sum_{h=1}^{H} (1 - \alpha) \left( \log(\breve{\Theta}_{m,h}^{\mathrm{PBB}}) - \log(\textstyle\sum_{h'=1}^{H} \breve{\Theta}_{m,h'}^{\mathrm{PBB}}) \right) \right) \\
&\quad + \sum_{h=1}^{H} \left( \log \frac{\Gamma(\sum_{l=1}^{L} \breve{B}_{l,h}^{\mathrm{PBB}})}{\prod_{l=1}^{L} \Gamma(\breve{B}_{l,h}^{\mathrm{PBB}})} \frac{\Gamma(\eta)^L}{\Gamma(L\eta)} + \sum_{l=1}^{L} \left( \breve{B}_{l,h}^{\mathrm{PBB}} - \eta \right) \left( \Psi(\breve{B}_{l,h}^{\mathrm{PBB}}) - \Psi(\textstyle\sum_{l'=1}^{L} \breve{B}_{l',h}^{\mathrm{PBB}}) \right) \right),
\end{aligned}
\tag{53}
$$

$$Q^{\mathrm{PBB}} = \left\langle \log \frac{q(\{\boldsymbol{z}^{(n,m)}\})}{p(\{\boldsymbol{w}^{(n,m)}\},\{\boldsymbol{z}^{(n,m)}\}|\boldsymbol{\Theta},\boldsymbol{B})} \right\rangle_{q^{\mathrm{PBB}}(\boldsymbol{\Theta},\boldsymbol{B},\{\boldsymbol{z}^{(n,m)}\})}$$

$$= -\sum_{m=1}^{M} N^{(m)} \sum_{l=1}^{L} V_{l,m} \log \left( \sum_{h=1}^{H} \frac{\breve{\Theta}_{m,h}^{\mathrm{PBB}}}{\sum_{h'=1}^{H} \breve{\Theta}_{m,h'}^{\mathrm{PBB}}} \frac{\exp\bigl(\Psi(\breve{B}_{l,h}^{\mathrm{PBB}})\bigr)}{\exp\bigl(\Psi(\sum_{l'=1}^{L} \breve{B}_{l',h}^{\mathrm{PBB}})\bigr)} \right). \tag{54}$$

Let us first consider the case when $\alpha < 1$. In this case, $F$ diverges to $F \to -\infty$ with fixed $N$, when $\breve{\Theta}_{m,h} = O(1)$ for any $(m,h)$ and $\breve{\Theta}_{m,h'} \to +0$ for all other $h' \neq h$. Therefore, the solution is sparse (so sparse that the estimator is useless).

When $\alpha \geq 1$, the solution satisfies the following stationary condition:

$$\breve{\Theta}_{m,h}^{\mathrm{PBB}} = \alpha - 1 + \sum_{n=1}^{N^{(m)}} \widehat{z}_h^{\mathrm{PBB}(n,m)}, \qquad \breve{B}_{l,h}^{\mathrm{PBB}} = \eta + \sum_{m=1}^{M} \sum_{n=1}^{N^{(m)}} w_l^{(n,m)} \widehat{z}_h^{\mathrm{PBB}(n,m)}, \tag{55}$$

$$\widehat{z}_h^{\mathrm{PBB}(n,m)} = \frac{\breve{\Theta}_{m,h}^{\mathrm{PBB}} \exp\left\{ \sum_{l=1}^{L} w_l^{(n,m)} \bigl( \Psi(\breve{B}_{l,h}^{\mathrm{PBB}}) - \Psi\bigl( \sum_{l'=1}^{L} \breve{B}_{l',h}^{\mathrm{PBB}} \bigr) \bigr) \right\}}{\sum_{h'=1}^{H} \breve{\Theta}_{m,h'}^{\mathrm{PBB}} \exp\left\{ \sum_{l=1}^{L} w_l^{(n,m)} \bigl( \Psi(\breve{B}_{l,h'}^{\mathrm{PBB}}) - \Psi\bigl( \sum_{l'=1}^{L} \breve{B}_{l',h'}^{\mathrm{PBB}} \bigr) \bigr) \right\}}. \tag{56}$$

In the same way as for VB and PBA learning, we can obtain the following lemma:

**Lemma 7** *Let* $\widehat{\boldsymbol{B}}^{\mathrm{PBB}} \widehat{\boldsymbol{\Theta}}^{\mathrm{PBB}\top} = \langle \boldsymbol{B}\boldsymbol{\Theta}^{\top} \rangle_{q^{\mathrm{PBB}}(\boldsymbol{\Theta},\boldsymbol{B})}$. *Then, it holds that*

$$\langle (\boldsymbol{B}\boldsymbol{\Theta}^{\top} - \widehat{\boldsymbol{B}}^{\mathrm{PBB}} \widehat{\boldsymbol{\Theta}}^{\mathrm{PBB}\top})_{l,m}^2 \rangle_{q^{\mathrm{PBB}}(\boldsymbol{\Theta},\boldsymbol{B})} = O_p(N^{-2}), \tag{57}$$

$$Q^{\mathrm{PBB}} = -\sum_{m=1}^{M} N^{(m)} \sum_{l=1}^{L} V_{l,m} \log(\widehat{\boldsymbol{B}}^{\mathrm{PBB}} \widehat{\boldsymbol{\Theta}}^{\mathrm{PBB}\top})_{l,m} + O_p(N^{-1}). \tag{58}$$

$Q^{\mathrm{PBB}}$ *is minimized when* $\widehat{\boldsymbol{B}}^{\mathrm{PBB}} \widehat{\boldsymbol{\Theta}}^{\mathrm{PBB}\top} = \boldsymbol{B}^* \boldsymbol{\Theta}^{*\top} + O_p(N^{-1})$, *and it holds that*

$$Q^{\mathrm{PBB}} = S + O_p(\widehat{J}N + LM).$$

$R^{\mathrm{PBB}}$ *is written as follows:*

$$R^{\mathrm{PBB}} = \left\{ MH(\alpha - 1) + \widehat{H}\left(L\eta - \tfrac{1}{2}\right) - \sum_{h=1}^{\widehat{H}} \left( \widehat{M}^{(h)}(\alpha - 1) + \widehat{L}^{(h)}\left(\eta - \tfrac{1}{2}\right) \right) \right\} \log N$$
$$+ (H - \widehat{H})\left(L\eta - \tfrac{1}{2}\right) \log L + O_p(H(M + L)). \tag{59}$$

Taking the different asymptotic limits, we obtain the following theorem:

**Theorem 4** *When* $\alpha < 1$, *each row vector of* $\widehat{\boldsymbol{\Theta}}^{\mathrm{PBB}}$ *has only one non-zero entry, and the PBB solution is sparse. Assume below that* $\alpha \geq 1$. *In the limit when* $N \to \infty$ *with* $L, M \sim O(1)$, *it holds that* $\widehat{J} = 0$ *with probability 1, and*

$$F^{\mathrm{PBB}} = S + \left\{ MH(\alpha - 1) + \widehat{H}\left(L\eta - \tfrac{1}{2}\right) - \sum_{h=1}^{\widehat{H}} \left( \widehat{M}^{(h)}(\alpha - 1) + \widehat{L}^{(h)}\left(\eta - \tfrac{1}{2}\right) \right) \right\} \log N$$
$$+ O_p(1).$$

*In the limit when* $N, M \to \infty$ *with* $\frac{M}{N}, L \sim O(1)$, *it holds that* $\widehat{J} = o_p(\log N)$, *and*

$$F^{\mathrm{PBB}} = S + \left\{ MH(\alpha - 1) - \sum_{h=1}^{\widehat{H}} \widehat{M}^{(h)}(\alpha - 1) \right\} \log N + o_p(N \log N).$$

*In the limit when* $N, L \to \infty$ *with* $\frac{L}{N}, M \sim O(1)$, *it holds that* $\widehat{J} = o_p(\log N)$, *and*

$$F^{\mathrm{PBB}} = S + HL\eta \log N + o_p(N \log N).$$

*In the limit when* $N, L, M \to \infty$ *with* $\frac{L}{N}, \frac{M}{N} \sim O(1)$, *it holds that* $\widehat{J} = o_p(N \log N)$, *and*

$$F^{\mathrm{PBB}} = S + H(M(\alpha - 1) + L\eta) \log N + o_p(N^2 \log N).$$

Theorem 4 states that the PBB solution is sparse when $\alpha < 1$. Below, we investigate the sparsity of the solution for $\alpha \geq 1$.

**In the limit when $N \to \infty$ with $L, M \sim O(1)$**

The coefficient of the leading term of the free energy is

$$\lambda^{\text{PBB}} = MH(\alpha - 1) + \sum_{h=1}^{\widehat{H}} \left( L\eta - \tfrac{1}{2} - \widehat{M}^{(h)}(\alpha - 1) - \widehat{L}^{(h)}\left(\eta - \tfrac{1}{2}\right) \right).$$

The solution is sparse if $\lambda^{\text{PBB}}$ is increasing of $\widehat{H}$, and dense if it is decreasing. We focus on the case when $\alpha \geq 1$. Eqs.(34)–(36) imply the following:

1. When $0 < \eta \leq \frac{1}{2L}$, the solution is sparse if

$$L\eta - \frac{1}{2} - \max_h M^{*(h)}(\alpha - 1) > 0, \text{ or equivalently, } \alpha < 1 - \frac{1}{\max_h M^{*(h)}}\left(\frac{1}{2} - L\eta\right),$$

   and dense if

$$\alpha > 1 - \frac{1}{\max_h M^{*(h)}}\left(\frac{1}{2} - L\eta\right).$$

   Therefore, the solution is always dense in this case.

2. When $\frac{1}{2L} < \eta \leq \frac{1}{2}$, the solution is sparse if

$$L\eta - \frac{1}{2} - \max_h M^{*(h)}(\alpha - 1) > 0, \text{ or equivalently, } \alpha < 1 + \frac{L\eta - \frac{1}{2}}{\max_h M^{*(h)}},$$

   and dense if

$$\alpha > 1 + \frac{L\eta - \frac{1}{2}}{\max_h M^{*(h)}}.$$

3. When $\eta > \frac{1}{2}$, the solution is sparse if

$$L\eta - \frac{1}{2} - \max_h \left( M^{*(h)}(\alpha - 1) + L^{*(h)}\left(\eta - \frac{1}{2}\right) \right) > 0, \tag{60}$$

   and dense if

$$L\eta - \frac{1}{2} - \max_h \left( M^{*(h)}(\alpha - 1) + L^{*(h)}\left(\eta - \frac{1}{2}\right) \right) < 0. \tag{61}$$

   Therefore, the solution is sparse if

$$L\eta - \frac{1}{2} - \max_h M^{*(h)}(\alpha - 1) - \max_h L^{*(h)}\left(\eta - \frac{1}{2}\right) > 0,$$

   or equivalently, $\alpha < 1 + \dfrac{1}{\max_h M^{*(h)}}\left( L\eta - \dfrac{1}{2} - \max_h L^{*(h)}\left(\eta - \dfrac{1}{2}\right) \right)$,

   and dense if

$$L\eta - \frac{1}{2} - \min_h M^{*(h)}(\alpha - 1) - \max_h L^{*(h)}\left(\eta - \frac{1}{2}\right) < 0,$$

   or equivalently, $\alpha > 1 + \dfrac{1}{\min_h M^{*(h)}}\left( L\eta - \dfrac{1}{2} - \max_h L^{*(h)}\left(\eta - \dfrac{1}{2}\right) \right)$.

   Thus, we can conclude that, in this case, the solution is sparse if

$$\alpha < 1 + \frac{L - 1}{2\max_h M^{*(h)}},$$

   and dense if

$$\alpha > 1 + \frac{L\eta - \frac{1}{2}}{\min_h M^{*(h)}}.$$

Summarizing the above, we have the following lemma:

**Lemma 8** *Assume that $\alpha \geq 1$. When $0 < \eta \leq \frac{1}{2L}$, the solution is always dense. When $\frac{1}{2L} < \eta \leq \frac{1}{2}$, the solution is sparse if $\alpha < 1 + \frac{L\eta - \frac{1}{2}}{\max_h M^{*(h)}}$, and dense if $\alpha > 1 + \frac{L\eta - \frac{1}{2}}{\max_h M^{*(h)}}$. When $\eta > \frac{1}{2}$, the solution is sparse if $\alpha < 1 + \frac{L - 1}{2\max_h M^{*(h)}}$, and dense if $\alpha > 1 + \frac{L\eta - \frac{1}{2}}{\min_h M^{*(h)}}$.*

**In the limit when** $N, M \to \infty$ **with** $\frac{M}{N}, L \sim O(1)$

The coefficient of the leading term of the free energy is given by

$$\lambda = M (H\alpha - 1) - \sum_{h=1}^{\widehat{H}} \widehat{M}^{(h)} (\alpha - 1) . \tag{62}$$

Although the predictive distribution does not necessarily converges to the true distribution, we can investigate the sparsity of the solution by considering the duplication rules (34)–(36) that keep $\widehat{B}\widehat{\Theta}^{\top}$ unchanged. It is clear that Eq.(62) is decreasing of $\widehat{H}$ if $\alpha > 1$. Combing this result with Theorem 4, which states that the PBB solution is sparse when $\alpha < 1$, and Lemma 8, we obtain the following corollary:

**Corollary 3** *Consider the limit when* $N \to \infty$ *with* $L, M \sim O(1)$. *When* $0 < \eta \leq \frac{1}{2L}$, *the PBB solution is sparse if* $\alpha < 1$, *and dense if* $\alpha > 1$. *When* $\frac{1}{2L} < \eta \leq \frac{1}{2}$, *the PBB solution is sparse if* $\alpha < 1 + \frac{L\eta - \frac{1}{2}}{\max_h M^{*(h)}}$, *and dense if* $\alpha > 1 + \frac{L\eta - \frac{1}{2}}{\max_h M^{*(h)}}$. *When* $\eta > \frac{1}{2}$, *the PBB solution is sparse if* $\alpha < 1 + \frac{L-1}{2\max_h M^{*(h)}}$, *and dense if* $\alpha > 1 + \frac{L\eta - \frac{1}{2}}{\min_h M^{*(h)}}$. *In the limit when* $N, M \to \infty$ *with* $\frac{M}{N}, L \sim O(1)$, *the PBB solution is sparse if* $\alpha < 1$, *and dense if* $\alpha > 1$.

### E.3  MAP Learning

The free energy for MAP learning is given as follows:

$$F^{\mathrm{MAP}} = \chi_\Theta + \chi_B + R^{\mathrm{MAP}} + Q^{\mathrm{MAP}}, \tag{63}$$

where $\chi_\Theta$ and $\chi_B$ are large constants corresponding to the negative entropies of the delta functions, and

$$
\begin{aligned}
R^{\mathrm{MAP}} &= \left\langle \log \frac{q(\boldsymbol{\Theta})q(\boldsymbol{B})}{p(\boldsymbol{\Theta}|\alpha)p(\boldsymbol{B}|\eta)} \right\rangle_{q^{\mathrm{MAP}}(\boldsymbol{\Theta},\boldsymbol{B})} \\
&= \sum_{m=1}^{M} \left( \log \frac{\Gamma(\alpha)^H}{\Gamma(H\alpha)} + \sum_{h=1}^{H} (1-\alpha) \left( \log(\breve{\Theta}_{m,h}^{\mathrm{MAP}}) - \log(\sum_{h'=1}^{H} \breve{\Theta}_{m,h'}^{\mathrm{MAP}}) \right) \right) \\
&\quad + \sum_{h=1}^{H} \left( \log \frac{\Gamma(\eta)^L}{\Gamma(L\eta)} + \sum_{l=1}^{L} (1-\eta) \left( \log(\breve{B}_{l,h}^{\mathrm{MAP}}) - \log(\sum_{l'=1}^{L} \breve{B}_{l',h}^{\mathrm{MAP}}) \right) \right),
\end{aligned} \tag{64}
$$

$$
\begin{aligned}
Q^{\mathrm{MAP}} &= \left\langle \log \frac{q(\{\boldsymbol{z}^{(n,m)}\})}{p(\{\boldsymbol{w}^{(n,m)}\},\{\boldsymbol{z}^{(n,m)}\}|\boldsymbol{\Theta},\boldsymbol{B})} \right\rangle_{q^{\mathrm{MAP}}(\boldsymbol{\Theta},\boldsymbol{B},\{\boldsymbol{z}^{(n,m)}\})} \\
&= -\sum_{m=1}^{M} N^{(m)} \sum_{l=1}^{L} V_{l,m} \log \left( \sum_{h=1}^{H} \frac{\breve{\Theta}_{m,h}^{\mathrm{MAP}}}{\sum_{h'=1}^{H} \breve{\Theta}_{m,h'}^{\mathrm{MAP}}} \frac{\breve{B}_{l,h}^{\mathrm{MAP}}}{\sum_{l'=1}^{L} \breve{B}_{l',h}^{\mathrm{MAP}}} \right).
\end{aligned} \tag{65}
$$

Let us first consider the case when $\alpha < 1$. In this case, $F$ diverges to $F \to -\infty$ with fixed $N$, when $\breve{\Theta}_{m,h} = O(1)$ for any $(h, m)$ and $\breve{\Theta}_{m,h'} \to +0$ for all other $h' \neq h$. Therefore, the solution is sparse (so sparse that the estimator is useless). Similarly, assume that $\eta < 1$. Then, $F$ diverges to $F \to -\infty$ with fixed $N$, when $\breve{B}_{l,h} = O(1)$ for any $(l, h)$ and $\breve{B}_{l',h} \to +0$ for all other $l' \neq l$. Therefore, the solution is useless.

When $\alpha \geq 1$ and $\eta \geq 1$, the solution satisfies the following stationary condition:

$$\breve{\Theta}_{m,h}^{\mathrm{MAP}} = \alpha - 1 + \sum_{n=1}^{N^{(m)}} \widehat{z}_h^{\mathrm{MAP}(n,m)}, \qquad \breve{B}_{l,h}^{\mathrm{MAP}} = \eta - 1 + \sum_{m=1}^{M} \sum_{n=1}^{N^{(m)}} w_l^{(n,m)} \widehat{z}_h^{\mathrm{MAP}(n,m)}, \tag{66}$$

$$\widehat{z}_h^{\mathrm{MAP}(n,m)} = \frac{\breve{\Theta}_{m,h}^{\mathrm{MAP}} \prod_{l=1}^{L} (\breve{B}_{l,h}^{\mathrm{MAP}})^{w_l^{(n,m)}}}{\sum_{h'=1}^{H} \left( \breve{\Theta}_{m,h'}^{\mathrm{MAP}} \prod_{l=1}^{L} (\breve{B}_{l,h'}^{\mathrm{MAP}})^{w_l^{(n,m)}} \right)}. \tag{67}$$

In the same way as for VB, PBA, and PBB learning, we can obtain the following lemma:

**Lemma 9** *Let* $\widehat{\boldsymbol{B}}^{\mathrm{MAP}}\widehat{\boldsymbol{\Theta}}^{\mathrm{MAP}\top} = \langle \boldsymbol{B}\boldsymbol{\Theta}^{\top}\rangle_{q^{\mathrm{MAP}}(\boldsymbol{\Theta},\boldsymbol{B})}$. *Then,* $Q^{\mathrm{MAP}}$ *is minimized when* $\widehat{\boldsymbol{B}}^{\mathrm{MAP}}\widehat{\boldsymbol{\Theta}}^{\mathrm{MAP}\top} = \boldsymbol{B}^{*}\boldsymbol{\Theta}^{*\top} + O_{p}(N^{-1})$, *and it holds that*

$$Q^{\mathrm{MAP}} = S + O_{p}(\widehat{J}N + LM).$$

$R^{\mathrm{MAP}}$ *is written as follows:*

$$R^{\mathrm{MAP}} = \left\{ MH\left(\alpha - 1\right) + \widehat{H}L\left(\eta - 1\right) - \sum_{h=1}^{\widehat{H}}\left(\widehat{M}^{(h)}\left(\alpha - 1\right) + \widehat{L}^{(h)}\left(\eta - 1\right)\right)\right\}\log N$$
$$+ (H - \widehat{H})L\left(\eta - 1\right)\log L + O_{p}(H(M + L)). \tag{68}$$

Taking the different asymptotic limits, we obtain the following theorem:

**Theorem 5** *When* $\alpha < 1$*, each row vector of* $\widehat{\boldsymbol{\Theta}}^{\mathrm{MAP}}$ *has only one non-zero entry, and the MAP solution is sparse. When* $\eta < 1$*, each column vector of* $\widehat{\boldsymbol{B}}^{\mathrm{MAP}}$ *has only one non-zero entry. Assume below that* $\alpha, \eta \geq 1$*. In the limit when* $N \to \infty$ *with* $L, M \sim O(1)$*, it holds that* $\widehat{J} = 0$ *with probability 1, and*

$$F^{\mathrm{MAP}} = S + \left\{ MH\left(\alpha - 1\right) + \widehat{H}L\left(\eta - 1\right) - \sum_{h=1}^{\widehat{H}}\left(\widehat{M}^{(h)}\left(\alpha - 1\right) + \widehat{L}^{(h)}\left(\eta - 1\right)\right)\right\}\log N$$
$$+ O_{p}(1).$$

*In the limit when* $N, M \to \infty$ *with* $\frac{M}{N}, L \sim O(1)$*, it holds that* $\widehat{J} = o_{p}(\log N)$*, and*

$$F^{\mathrm{MAP}} = S + \left\{ MH\left(\alpha - 1\right) - \sum_{h=1}^{\widehat{H}}\widehat{M}^{(h)}\left(\alpha - 1\right)\right\}\log N + o_{p}(N\log N).$$

*In the limit when* $N, L \to \infty$ *with* $\frac{L}{N}, M \sim O(1)$*, it holds that* $\widehat{J} = o_{p}(\log N)$*, and*

$$F^{\mathrm{MAP}} = S + HL(\eta - 1)\log N + o_{p}(N\log N).$$

*In the limit when* $N, L, M \to \infty$ *with* $\frac{L}{N}, \frac{M}{N} \sim O(1)$*, it holds that* $\widehat{J} = o_{p}(N\log N)$*, and*

$$F^{\mathrm{MAP}} = S + H(M(\alpha - 1) + L(\eta - 1))\log N + o_{p}(N^{2}\log N).$$

Theorem 5 states that the MAP solution is sparse when $\alpha < 1$. However it provides no information on the sparsity of the MAP solution for $\eta < 1$. Below, we investigate the sparsity of the solution for $\alpha, \eta \geq 1$.

**In the limit when** $N \to \infty$ **with** $L, M \sim O(1)$

The coefficient of the leading term of the free energy is

$$\lambda^{\mathrm{MAP}} = MH\left(\alpha - 1\right) + \sum_{h=1}^{\widehat{H}}\left(L(\eta - 1) - \widehat{M}^{(h)}\left(\alpha - 1\right) - \widehat{L}^{(h)}\left(\eta - 1\right)\right).$$

The solution is sparse if $\lambda^{\mathrm{MAP}}$ is increasing of $\widehat{H}$, and dense if it is decreasing. We focus on the case when $\alpha, \eta \geq 1$. Eqs.(34)–(36) imply the following:

The solution is sparse if

$$L(\eta - 1) - \max_{h}\left(M^{*(h)}\left(\alpha - 1\right) + L^{*(h)}\left(\eta - 1\right)\right) > 0, \tag{69}$$

and dense if

$$L(\eta - 1) - \max_{h}\left(M^{*(h)}\left(\alpha - 1\right) + L^{*(h)}\left(\eta - 1\right)\right) < 0. \tag{70}$$

Therefore, the solution is sparse if

$$L(\eta - 1) - \max_{h}M^{*(h)}\left(\alpha - 1\right) - \max_{h}L^{*(h)}\left(\eta - 1\right) > 0,$$

$$\text{or equivalently,} \ \ \alpha < 1 + \frac{(L - \max_{h}L^{*(h)})(\eta - 1)}{\max_{h}M^{*(h)}},$$

ad dense if

$$L(\eta - 1) - \min_h M^{*(h)} (\alpha - 1) - \max_h L^{*(h)} (\eta - 1) < 0,$$

or equivalently, $\alpha > 1 + \dfrac{(L - \max_h L^{*(h)})(\eta - 1)}{\min_h M^{*(h)}}.$

Thus, we can conclude that the solution is sparse if

$$\alpha < 1,$$

and dense if

$$\alpha > 1 + \frac{L(\eta - 1)}{\min_h M^{*(h)}}.$$

Summarizing the above, we have the following lemma:

**Lemma 10** *Assume that $\eta \geq 1$. The solution is sparse if $\alpha < 1$, and dense if $\alpha > 1 + \frac{L(\eta-1)}{\min_h M^{*(h)}}$.*

**In the limit when $N, M \to \infty$ with $\frac{M}{N}, L \sim O(1)$**

The coefficient of the leading term of the free energy is given by

$$\lambda^{\mathrm{MAP}} = MH (\alpha - 1) - \sum_{h=1}^{\widehat{H}} \widehat{M}^{(h)} (\alpha - 1). \tag{71}$$

Although the predictive distribution does not necessarily converges to the true distribution, we can investigate the sparsity of the solution by considering the duplication rules (34)–(36) that keep $\widehat{B}\widehat{\Theta}^{\top}$ unchanged. It is clear that Eq.(71) is decreasing of $\widehat{H}$ if $\alpha > 1$. Combing this result with Theorem 5, which states that the MAP solution is sparse if $\alpha < 1$, and Lemma 10, we obtain the following corollary:

**Corollary 4** *Assume that $\eta \geq 1$. In the limit when $N \to \infty$ with $L, M \sim O(1)$, the MAP solution is sparse if $\alpha < 1$, and dense if $\alpha > 1 + \frac{L(\eta-1)}{\min_h M^{*(h)}}$. In the limit when $N, M \to \infty$ with $\frac{M}{N}, L \sim O(1)$, the MAP solution is sparse if $\alpha < 1$, and dense if $\alpha > 1$.*

### E.4 Summary of Results

Summarizing Corollary 1, Corollary 2, Corollary 3, and Corollary 4 completes the proof of Theorem 2. $\qquad\square$