[Reviews · NeurIPS 2014]

Submitted by Assigned_Reviewer_12

In this paper, authors analyze sparsity of the posterior parameters in LDA using a variational Bayesian algorithm. They derive an expression for the VB free energy which shows its asymptotic behaviour with respect to number of words (N), number of documents (M), vocabulary size (L) etc. Their results suggest that, for certain settings of L,M,N, the sparsity behaviour changes drastically at a particular hyper-parameter setting. These changes differ from those of MAP and partial-Bayes algorithms.

The problem discussed in this paper is original, interesting, and is perhaps useful too. The arguments are supported by rigorous theorems. I found the paper to be really hard to read and understand. The paper is not written clearly, and there are many steps in the derivations that are not outlined. Therefore, I was unable to understand the derivation the theorem. I have few suggestions below to improve this part. Another big problem that I find with this work is the context. I think authors need to discuss the significance of their results in the context of real data. Perhaps, a more extensive experiment section will be useful. I have some suggestions on this below.

Detailed comments:

The paper discusses, at several occasion (e.g. second paragraph), sparsity in general latent variable models (e.g. mixture model etc.). If the notion of sparsity is comparable in these models, then it will be good to include a general definition of sparsity in latent variable models. Otherwise, it will be good to include a discussion of what exactly is meant by weaker or stronger sparsity in these models, e.g. mixture model, and how does it compare to sparsity in LDA.

The LDA notation used in the paper is not-standard and makes it difficult to read for people familiar with LDA. I highly recommend to change the notation. Also, please include a graphical model in Section 2.1. Use q for approximate distribution instead of r.

Paragraph at line 185 is not clear. It is perhaps clearer to state that you assume A* and B* of rank K* that can model the data well.

Define O_p for people who are not familiar with it.

Explain more clearly the average number of nonzero entries in line 201.

In Line 253, you say "Note that Theorem 1 combined with Lemma 1 implies the consistency". This isn't trivial (at least to me), although I can see how to get there. It is perhaps a good idea to write these results as theorem rather than the Lemma that you have. At least it gives a clear statement to the reader about what is being proved in the paper. You should then include a detailed proof (really detailed) in the Appendix. I will suggest the same for Corollary 1 and 2. I do not see how you arrive at these results so trivially. It will be good have details on this in the Appendix.

The proofs in Appendix need to be more detailed too. Currently, there are statements that are not easy to derive, or at least I could not derive them myself. E.g. line 511, "because of the independence between A and B, the posterior distribution must converge to a point... ". I don't know why? In line 522, "we can show that...". Unfortunately, I wasn't able to show this, so some help will make the experience for readers less frustrating. It was so discouraging that I couldn't make it beyond the first page of Appendix.

Please provide some context for Corollary 1 and 2. You have discussed two settings of L,M,N where you have clear statements. What are the significance of these cases? when they are likely to occur? do real data have one setting more than others? If we only know about sparsity of B, can we still conclude something general about the whole situation?

Please make a table comparing PBB, PBA, MAP and VB vs. settings of c_a and c_b. This will improve clarity in summarizing the results. It seems right now that integrating A out gives the sparsity at c_a=1/2 behaviour. Do you think that will be true in general?

Line 305, MAP tends to induce stronger sparsity. Please clarify this further. I assume you mean that VB does not include c_a >1/2 cases therefore will have less sparsity?

Line 323. L and M would be comparable to N in standard text analysis. I don't think it's correct to say that.

Results section. Why don't you include other text datasets on which LDA has been applied more frequently rather than LastFM? Also provide a reference that applies LDA on lastFM.

Please include more datasets. Remove c_b from plots if it doesn't affect the sparsity. Rather include more datasets and more different scenarios for different values of L,M, and N.

Summary: The problem discussed in this paper is original, interesting, and is perhaps useful too. The arguments are supported by rigorous theorems. However, I found the paper very difficult to follow and because of that I was also unable to check the proofs of theorems. If written clearly, this paper could be useful.

-----------
Added after the author feedback.
-----------
Thanks to the authors for their response. I am sure that this paper is useful to the community. I could not verify the theorems presented in the paper since the paper was difficult to read, but I do hope that the authors are going to make an attempt in improving clarity of the paper.

Submitted by Assigned_Reviewer_25

The paper theoretically proofs the resulting not so sparse characteristics of using variational Bayesian learning on a LDA model.

The paper is well written and clear. The math is solid, however the paper lacks a solid results section. The results provided shown that the solutions learned have indeed the characteristics that the authors predicted with mathematical proofs, but the results do not shown any actual performance advantages over other possible solutions which then limits its significance overall
Summary: Hard to measure the impact of the paper due to a weak results section. Comparisons on the results using other techniques should be added

Submitted by Assigned_Reviewer_27

The paper presents a theoretical study of variational Bayes (VB) for the LDA topic model under asymptotic settings. The authors show that MAP estimation (under assumptions on data sizes) results in stronger sparse solutions than VB. Further, they show the consistency of VB for LDA. A study on a collaborative filtering data set shows that the sparsity (measured as avg. number of non-zero entries in document topic-proportions) varied as a function the sparsity threshold (the hyperparameter on per-document topic proportions) provides empirical evidence for the theory. Specifically, the threshold is larger in MAP than VB.

- The strength of this paper is a rigorous study of an important practical problem: why does sparsity behavior drastically change in LDA with changes to hyperparameters.

- One drawback of the method, that the authors acknowledge, is the theory does not provide insights when the number of documents is much smaller than #words and vocabulary size. This is probably not a major concern, as in standard text analysis the number of documents is larger/same order as #words/vocabulary size.

- My main complaint is the paper makes little effort to identify why this sparsity behavior in LDA is opposite to several other models where generally VB induces greater sparsity than MAP. Given the tools they develop, it's curious why this wasn't explored for a closely related model (such as Poisson matrix factorization).

Two issues here:

1. The authors conjecture that in these models the sparsity threshold is mainly a function of the degree of freedom of a single component, whereas in LDA a newly added topic explains a large number of documents, discounting the effect of model complexity. However, is this not true of fully-observed matrix factorization? A newly added components serves to explain a large number of user preferences and item attributes.

2. It would have been interesting to compare to closely-related matrix factorization models (such as Poisson factorization) or Poisson MRF (Inouye et al. 2014).

- My other concern with this paper is that the chosen real-world data is not a large corpus but a small collaborative filtering Last.fm data set consisting of users rating songs. Why? (I tried to follow the last paragraph of the discussion, which might answer this. But it's not clear to me.) Further, only a single real data set is studied.

minor:

- I found the notation difficult to follow. It would have been useful to use the notation from the LDA paper, given it's wide use in literature.
Summary: A solid theoretical study of an important practical problem: why does sparsity behavior drastically change in LDA with changes to hyperparameters, and how doe VB compare to MAP in inducing sparsity. The paper makes related contributions as well, showing the consistency of VB-LDA. Somewhat narrow numerical results support theory. Significant concerns around (a) why a text data set was not studied, and (b) why the paper does not study why closely related models, such as fully-observed matrix factorization, that exhibit different sparsity behavior.
Author Feedback
Author rebuttal: Dear Reviewers,

Thank you very much for your sensible comments on our manuscript.
Please find our answers to the questions below.

With best regards,
Authors

Reply to Reviewer 12:

1. Definition of sparsity.
Thanks for your suggestion.
Our goal is to clarify if VB exhibits the automatic relevance determination (ARD) effect,
which automatically prunes irrelevant degrees of freedom.
To this end, assuming that the true number K* of topics (or rank) is small,
we investigated if the solution is expressed with K* topics or with more topics.
If the solution is expressed with K* topics, we say the solution is sparse.
In this case, the ARD effect provides the correct estimator for the relevant number of topics.
On the other hand, if the solution is expressed with H = min(L,M) topics, we say the solution is dense.

In the original manuscript, this point was unclear partly because we used \hat{H} and \hat{L}, defined in Line 203,
as criteria of sparsity. However, \hat{H} and \hat{L} depend on the sparsity of the true signal matrix, and do not directly reflect the ARD behavior.
In the revision, we will change the criterion for the sparsity to the estimated number of topics (or rank), i.e.,
\hat{K} = sum_{h=1}^H \theta(\sum_{m=1}^M A_{m, h} / (M \sum_{h=1}^H A_{m, h}) \sim O_p(1)),
in theory and experiments.
Whether \hat{K} is close to K* or close to H=min(L,M) indicates whether the ARD effect correctly prunes irrelevant degrees of freedom or not.

Using \hat{K} as a criterion of sparsity is consistent with previous works on other latent variable models, where the number of components with non-zero weights in mixture models [20,21],
and the estimated rank in matrix factorization (MF) [15,16] were investigated.
We will clarify this point in the revision.

2. Strength of sparsity.
We say that an inference method X exhibits weaker sparsity than another method Y
if X requires a sparser prior to provide a sparse solution than Y does.
c_a and c_b directly control the sparsity of the Dirichlet prior (the smaller c_a is, the sparser the prior of A is).
Therefore, because VB requires smaller c_a to make the estimator (the mean of the posterior) sparse, VB is shown to exhibit weaker sparsity than MAP.
On the other hand, in mixture models, VB provides a sparse solution with a less sparse prior than MAP [20], and therefore VB is shown to exhibit stronger sparsity than MAP.

3. Statements, derivations, and proofs of theoretical results.
We will give a clear statement on the consistency as a theorem,
and much more detailed proofs of lemmas, theorems, and corollaries in Appendices.
We apologize that the proofs in the original manuscript were hard to read.
We will give much effort to make them easy to read for most readers.

4. Significance of our results. Why Last.FM dataset was chosen to show the validity of our theory.
Our original motivation was to clarify the leading term of the free energy,
hoping that it would give information on the sparsity of the solution,
and hence on whether the ARD effect correctly prunes irrelevant degrees of freedom.
As mentioned in Line 269, it however turned out that
the leading term explains the sparsity of the solution only when L << N.

This is why we did not use a large text corpus, which typically has L comparable to or larger than N
and thus our theory does not inform the sparsity of the solution.
On the other hand, in human activity data like Last.FM,
the number N of activities per user can be much larger than the number L of focused artists.
For this reason, we used the small scale Last.FM data to validate our theory.

5. Notation.
We will adopt the standard notation for LDA, used in [5],
and include its graphical model in Section 2.1.
We will give the definition of O_p.

6. Paragraph 185.
We will revise the manuscript as the reviewer suggests.

7. Line 201.
As mentioned above, we will change the criterion,
and make the meaning of the criterion clear.

8. Reference on LDA for Last.FM.
We will cite references where LDA is used for analyzing the Last.FM data.

9. Comparison with a table.
We will give a table to summarize our theory.
As long as our assumption holds, we conclude that
integrating A out gives the sparsity around c_a = 1/2.

10. We will remove c_b from the plots, and include more different scenarios in the experiment.

Reply to Reviewer 25:

1. Significance of the paper.
In this paper, we do not propose a new method,
but analyze the behavior of existing methods.

There is a paper, which we should have cited:
A. Asuncion, M. Welling, P. Smyth, and Y. W. Teh,
"On smoothing and inference for topic models," UAI2009.
There, the update rules of VB and MAP are compared,
based on the approximation exp(psi(n)) \approx n - 0.5,
and the behavior of the prediction errors is experimentally investigated.
We see our theory as a first step to theoretically explain their results
by investigating the sparsity, which is often related to the prediction accuracy.

2. Experimental result.
Because our goal is to explain the behavior of existing methods,
we do not show advantages of any method.
We performed experiments only for validating our theory.

Reply to Reviewer 27:

1. We have not really understood why the sparsity behavior in LDA is opposite to other models.
As the reviewer pointed out, our explanation (the degrees of freedom are shared over multiple users) also applies to the fully-observed MF, where VB was shown to exhibit stronger sparsity than MAP in [15]. Further investigation is necessary, and
we appreciate reviewer's suggestion to investigate Poisson MF,
which might be in between LDA and Gaussian MF.
Analyzing Poisson MRF can also be interesting future work.

2. Why small scale collaborative filtering data are used in the experiment?
Please see Answer 4. to Reviewer 12.